

SciPost Phys. Lect. Notes 53 (2022)

# Les Houches Lectures on Indirect Detection of Dark Matter

**Tracy R. Slatyer⋆**

Center for Theoretical Physics,
Massachusetts Institute of Technology,
Cambridge, MA 02139, USA

⋆ tslatyer@mit.edu

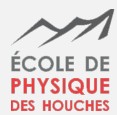

*Part of the Dark Matter*
*Session 118 of the Les Houches School, July 2021*
*published in the Les Houches Lecture Notes Series*

## Abstract

These lectures, presented at the 2021 Les Houches Summer School on Dark Matter, provide an introduction to key methods and tools of indirect dark matter searches, as well as a status report on the field circa summer 2021. Topics covered include the possible effects of energy injection from dark matter on the early universe, methods to calculate both the expected energy distribution and spatial distribution of particles produced by dark matter interactions, an outline of theoretical models that predict diverse signals in indirect detection, and a discussion of current constraints and some claimed anomalies. These notes are intended as an introduction to indirect dark matter searches for graduate students, focusing primarily on intuition-building estimates and useful concepts and tools.

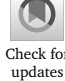

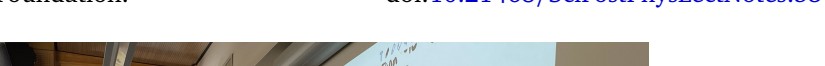

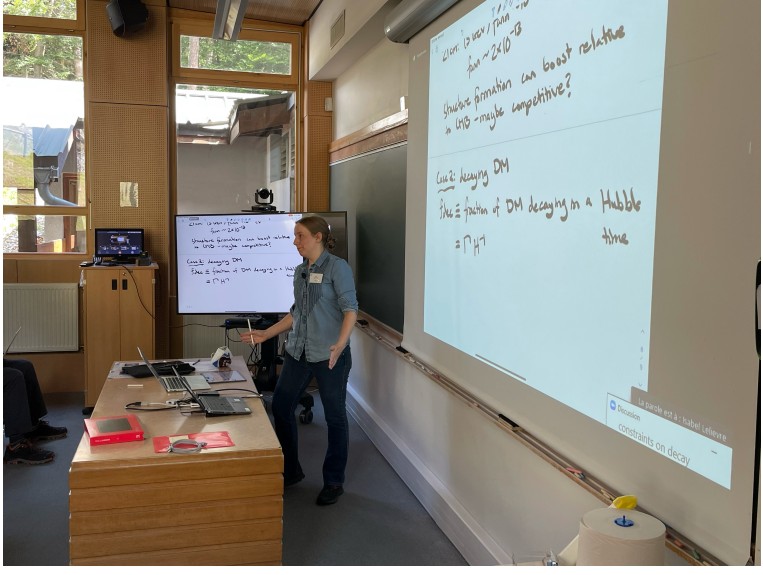

# 1  Lecture 1: introduction to indirect detection and signals through cosmic time

## 1.1  Introduction

The dark matter (DM) searches falling under the umbrella of *indirect detection* seek to identify possible visible products of DM interactions, originating from the DM already present in the cosmos. In particular, indirect searches often focus on searching for Standard Model (SM) particles produced from DM through decays, annihilations, oscillations, or other mechanisms, or the secondary effects of those particles.

Indirect detection benefits from the huge amount of ambient DM (with energy density five times that of baryonic matter, over cosmological volumes), and the existence of telescopes – originally designed to answer other questions in astronomy and astrophysics – that provide sensitivity to exotic sources of SM particles, especially photons, over an enormous range of energies. However, indirect searches face challenges because DM is known to interact only weakly with the SM, so the rate of particle production is expected to be small, and many possible detection channels have large potential backgrounds from astrophysical particle production.

Nonetheless, indirect detection covers an enormous range of detection channels and target regions, and consequently searches falling under the indirect-detection umbrella can have sensitivity to models and physics questions that are difficult or impossible to probe in Earth-based experiments. Some of the classic questions include the lifetime of DM (longer than the age of the universe, but perhaps not infinite), and the origin of DM in models where the abundance is fixed through annihilation. Observations of compact objects (supernovae, neutron stars, black holes, exoplanets, etc) and the very early universe can probe new physics in conditions of density, temperature, and electromagnetic fields that are not readily available on Earth; in recent years there has been considerable work on using such observations to probe DM and other new physics.

## 1.2  Simple estimates for dark matter annihilation over the history of the cosmos

As a starting point, let us seek to understand how annihilating or decaying DM might leave signatures in various astrophysical and cosmological observables. Let us begin by examining the fraction of DM that would annihilate per Hubble time, over the history of the cosmos, if DM was a thermal relic with $s$-wave annihilation as a benchmark (see the appendices for a review of thermal relic DM if needed). Here we will take $a(t)$ to denote the scale factor for expansion, so the Hubble parameter is $H = (1/a)da/dt$ and a Hubble time is $H^{-1}$.

After freezeout, the DM number density $n$ scales as $n \propto a^{-3}$, and the annihilation rate scales as density squared. The fraction of DM that annihilates in a Hubble time is thus given approximately by:

$$f_{\text{ann}} \approx \frac{n^2 \langle \sigma v_{\text{rel}} \rangle}{n} H^{-1} \propto a^{-3} \langle \sigma v_{\text{rel}} \rangle H^{-1}, \tag{1}$$

where $\langle \sigma v_{\text{rel}} \rangle$ is the velocity-weighted average DM annihilation cross section and $v_{\text{rel}}$ denotes the relative velocity between DM particles. Here we are assuming DM is its own antiparticle, but the scaling relations we derive here do not change if the DM has an antiparticle with equal abundance.

During radiation domination, $H^2 \propto \rho \propto a^{-4}$, so $H \propto a^{-2}$ and $f_{\text{ann}} \propto a^{-1}$ (assuming $\langle \sigma v_{\text{rel}} \rangle$ is constant, which is true for $s$-wave annihilation of non-relativistic particles to much lighter species). During matter domination, $H^2 \propto \rho \propto a^{-3}$, so $f_{\text{ann}} \propto a^{-1.5}$. During dark energy domination, $H$ is independent of $a$, and so $f_{\text{ann}} \propto a^{-3}$.

At freezeout, by definition $n \langle \sigma v_{\text{rel}} \rangle \sim H$, i.e. a $\mathcal{O}(1)$ fraction of DM particles annihilates per Hubble time, and so we have $f_{\text{ann}} \sim 1$. Using these scaling relations, we can quickly estimate what fraction of the DM is annihilating at later times in the universe's history. We generally expect freezeout to occur before Big Bang nucleosynthesis (BBN) to avoid disrupting nucleosynthesis, which implies a freezeout temperature of order 1 MeV at the lowest (it may be much higher); matter-radiation equality occurs at a temperature around 1 eV. Thus we expect $f_{\text{ann}}$ to drop to $< 10^{-6}$ by matter-radiation equality. This is an upper bound; the true value may be many orders of magnitude smaller. Today the temperature of the universe is a few $\times 10^{-4}$ eV, and most of the remaining expansion (about a factor of 3000 in $a$) has occurred during matter domination, meaning $f_{\text{ann}}$ has decreased by a further factor of about $6 \times 10^{-6}$. Thus we expect the fraction of DM that annihilates per Hubble time to be of order $10^{-11}$ at the very most, in regions where the DM density takes its cosmological average value. In particular, this means the modification to the overall DM density after freezeout is minuscule, unless there is an effect that greatly amplifies the annihilation rate at late times; to get a $\mathcal{O}(1)$ change to the local DM abundance from annihilation, we would need a density 11 orders of magnitude higher than the cosmological density. For comparison, the DM density in the neighborhood of the Earth is roughly a factor of $4 \times 10^5$ higher than the cosmological density, so even at this upper bound (which we will see is strongly excluded), we would expect only about one in $10^6$ DM particles to annihilate per Hubble time.

However, even such a small fraction of annihilating DM can have very marked effects on the history of the cosmos. If we are interested in the effects on ordinary matter from DM annihilation, it is often helpful to examine the energy liberated by DM annihilation per baryon per Hubble time. At late times there is roughly 5 GeV of energy stored in DM for every baryon (since the mass of a proton/neutron is roughly 1 GeV), and this ratio is fixed, so we can just multiply $f_{\text{ann}}$ by 5 GeV to obtain the energy injection per baryon in a Hubble time, $\varepsilon \sim 5\text{GeV} f_{\text{ann}} \sim (5\text{GeV}/T_{\text{freezeout}}) T_{\text{today}}$, where the last approximate equality holds during the radiation epoch (ignoring changes in the number of relativistic degrees of freedom, by assuming temperature scales as $1/a$).

This means that for 100 GeV thermal relic DM, which freezes out at a temperature around 5 GeV, the energy injection per baryon in a Hubble time is comparable to the temperature of the cosmic microwave background (CMB). Down to around redshift 200 (temperatures of about 0.1 eV), the CMB temperature is equal to the baryon temperature – so this means DM annihilation is expected to inject enough energy in every Hubble time to change the kinetic energy of all baryons in the universe by a $\mathcal{O}(1)$ amount throughout the radiation-dominated epoch, for 100 GeV thermal relic DM. (In practice, this effect does not literally change the temperature of the baryons by this factor, as the CMB is tightly coupled to the baryons and can act as a heat sink.) Lighter DM, that freezes out later, will inject even more energy.

BBN occurs at a temperature of around 1 MeV; this calculation indicates that 100 GeV thermal relic DM will inject roughly 1 MeV of energy for every baryon in the universe in a Hubble time during the BBN epoch. This amount of energy injection has the potential to affect subdominant nuclear abundances during nucleosynthesis (see e.g. Ref. [1] for a review, and Ref. [2] for a more recent analysis of some aspects of the problem). Note that this is distinct from the BBN constraint based on the number of relativistic degrees of freedom, which constrains light DM with masses $\lesssim$ 1 MeV (e.g. [3]).

The next important observable epoch is that of recombination, when the CMB radiation is released, the temperature of the photon bath is around 0.2 eV, and the universe has recently become matter-dominated. Our quick estimate above suggests that annihilation of 100 GeV thermal relic DM should release roughly 0.2 eV per baryon in a Hubble time during the recombination epoch. Since the ionization energy of hydrogen is 13.6 eV, this means such annihilations have the power to ionize roughly 1-2% of the hydrogen in the universe!

Recombination is characterized by a sharp drop in the ambient ionization level and a corresponding increase in the amount of neutral hydrogen; an increase in the post-recombination ionization level by 1-2% would be very visible, as the extra free electrons would provide a screen to the photons of the CMB. Measurements of the CMB are currently sensitive to changes of just a few $\times 10^{-4}$ in the ionization fraction during the cosmic dark ages after recombination (e.g. Ref. [4]). If all the energy from DM annihilation were to go into ionization, this implies we could test any models with $5\,\mathrm{GeV} f_{\mathrm{ann}} \gtrsim 13.6\,\mathrm{eV} \times \mathrm{few} \times 10^{-4}$, i.e. $f_{\mathrm{ann}} \gtrsim 10^{-12}$, corresponding to freezeout temperatures $\lesssim$ 200 GeV.

A more careful calculation, taking into account the presence of recombination as well as ionization, and the fact that not all the injected power goes into ionization, implies the limit on $f_{\mathrm{ann}}$ is closer to $f_{\mathrm{ann}} \lesssim 10^{-11}$ (or weaker if there is a large branching ratio into neutrinos or other states that do not interact electromagnetically). Including the temperature changes of the photon bath due to the changing number of relativistic degrees of freedom, and the evolution of the DM annihilation rate after matter-radiation inequality, somewhat lowers the expected value of $f_{\mathrm{ann}}$ for thermal relic DM compared to this crude first-pass estimate (one can also just use the thermal relic annihilation cross section computed numerically, rather than the rough approximations above). The net effect is that thermal relic DM with a velocity-independent $\sigma v$ can be ruled out below mass scales of $\mathcal{O}(10-30)$ GeV, depending on the annihilation channel [5].

One might also ask about constraints on the temperature of the baryons. After recombination, energy injected by DM annihilation/decay is partitioned between ionization, heating, and modifications to the free-streaming photon spectrum, so heating and ionization constraints can be complementary. The limiting factor here is that at redshifts higher than $z \sim 200$, the CMB and baryons maintain the same temperature, and temperature increases to the baryons are redistributed to the (much more abundant) CMB photons. On the other hand, at redshifts $z \sim 6-10$, the temperature of the universe is expected to increase rapidly due to photoheating from stars; it is still possible to set limits on DM annihilation and decay from $z \lesssim 6$, but the temperature change that can be tested is much larger, due to the high baseline temperature.

At present, observations of the Lyman-$\alpha$ forest provide constraints on the gas temperature around $T \sim 10^4 K$ for $z \sim 2-6$ [6,7]; in future, observations of primordial 21cm radiation may provide measurements of the gas temperature prior to reionization, which could be as low as $\mathcal{O}(10)$ K in the standard cosmological model (as discussed in e.g. Ref. [8], which provides a general review of 21cm cosmology). We will later briefly discuss a claim of detection of primordial 21cm radiation by the EDGES experiment, which if confirmed could suggest an even lower gas temperature.

These temperatures correspond to kinetic energy per baryon of $\sim$ 1 eV and $\sim 10^{-3}$ eV respectively; if we assume this energy injection takes place over a Hubble time, and that these

probes would have sensitivity to $\mathcal{O}(1)$ changes in the gas temperature, we can estimate they would have sensitivity to $f_{\mathrm{ann}} \sim 1\mathrm{eV}/5\mathrm{GeV} \sim 2 \times 10^{-10}$ and $f_{\mathrm{ann}} \sim 10^{-3}\mathrm{eV}/5\mathrm{GeV} \sim 2 \times 10^{-13}$ respectively, at timescales of order 100 million to 1 billion years ($\sim 10^{15} - 10^{16}$ s). Because these measurements would correspond to redshifts a factor of $\sim 100$ lower than the CMB, the naive scaling of $f_{\mathrm{ann}}$ with $a$ suggests they would not be competitive with the CMB bounds; however, DM structure formation at late times may change this conclusion especially for 21cm observations.

Given the existing CMB limits, we can see that our previous upper limit of $\sim 10^{-11}$ of DM annihilating today (in regions of cosmological average density), corresponding to a freezeout temperature of 1 MeV for a standard thermal relic, was far too high; we would have observed an enormous signal in the CMB if that were the case! Taking the freezeout temperature to be 5 GeV instead of 1 MeV, as appropriate for 100 GeV DM, we would need to lower our estimate of the upper bound by a factor of 5000; thus outside bound structures, in the present day, one would expect only a few in $\sim 10^{15}$ DM particles to annihilate in a Hubble time. In the neighborhood of the Earth, we might expect a few $\times 10^{-10}$ DM particles to annihilate per Hubble time.

(We can alternatively simply calculate $n\langle\sigma v_{\mathrm{rel}}\rangle$ for a number density of $(0.4\,\mathrm{GeV}/m_{\mathrm{DM}})/\mathrm{cm}^3$ and a cross section of $2\times 10^{-26}$ cm$^3$/s, and compare to the lifetime of the universe, $\tau \sim 4\times 10^{17}$ s; this gives us a rate of a few $\times 10^{-11}$ particles annihilating per Hubble time in the neighborhood of the Earth, for 100 GeV DM.)

Clearly, DM annihilation is not expected to deplete the DM content of our Galactic halo anytime soon! This calculation is also illustrative for understanding the self-interaction cross sections required for DM-DM scatterings to impact the small-structure of halos (e.g. Ref. [9]); typically, an average DM particle in the halo must interact at least once in the dynamical time of the halo for self-interactions to have a substantial effect. From the calculation above, we see that even if the dynamical time of the halo approaches the Hubble time, the required self-interaction cross section would need to be 10-11 orders of magnitude above the thermal relic cross section, for our benchmark of 100 GeV DM.

## 1.3 Simple estimates for dark matter decay over the history of the cosmos

For decaying DM, we can perform very similar estimates. The fraction of DM decaying per unit time is just the decay width $\Gamma$, so the fraction of DM decaying in a Hubble time is $f_{\mathrm{dec}} = \Gamma H^{-1}$. This quantity scales with redshift just as $H^{-1}$, and so satisfies $f_{\mathrm{dec}} \propto a^2$ during radiation domination, $f_{\mathrm{dec}} \propto a^{1.5}$ during matter domination, and $f_{\mathrm{dec}} =$ constant during dark energy domination. Consequently, decaying DM signals tend to be dominated by low redshifts / late times. For example, BBN occurs when the universe is about one second old, so for DM with a lifetime comparable to that of the universe, $\sim 10^{18}s$, the decaying fraction is about $10^{-18}$ – compare to the case of 100 GeV annihilating thermal relic DM, where it was about 1/5000! The CMB constraints discussed above, which test a fraction of DM converting to SM particles of around $10^{-11}$ (per Hubble time) when the lifetime of the universe was around a million years or $\sim 10^{14}$ s, thus test decay lifetimes of around $10^{25}$ s.

This argument suggests that it will be advantageous to look for late-time signals for decaying DM models. Consider the estimated limits on heating from Lyman-$\alpha$ and (in future) 21cm observations: we estimated these would have sensitivity to $f_{\mathrm{dec}} \sim 2 \times 10^{-10}$ (Lyman-$\alpha$) and in future $f_{\mathrm{dec}} \sim 2 \times 10^{-13}$ (21cm) during epochs where $H^{-1} \sim 10^{15-16}s$. This suggests Lyman-$\alpha$ observations could have sensitivity to decay lifetimes of $\mathcal{O}(10^{25-26})s$, comparable to the CMB bounds, and future 21cm observations of the late cosmic dark ages could potentially probe lifetimes around $10^{28-29}$ s.

Of course, we are not restricted to only cosmological probes. We will see that searches for DM decays in regions of high DM density can currently set lifetime bounds around $10^{27-28}$

seconds, corresponding to $f_{\text{dec}} \sim 10^{-10}$ in the present day.

Furthermore, these limits do not apply only to complete decay of particle DM. If a DM particle de-excites from a more massive state to a lighter state, we can repeat the same estimate but taking into account that only a fraction of the energy stored in the DM particle's mass is converted to SM particles. In particular, this applies to decay of primordial black holes via emission of Hawking radiation (see lectures by Prof. Carr and Dr. Kuhnel): black holes with the right mass range can serve as a viable DM candidate, and the conversion of their mass into visible particles via Hawking radiation can be treated just like decaying particle DM.

Finally, there are bounds on DM decaying even into invisible particles, from purely gravitational effects. However, these bounds generally only constrain lifetimes up to 1 order of magnitude longer than the age of the universe (e.g. [10,11]); being able to observe the visible products of annihilation (or their secondary effects) improves the limits by $\sim 8 - 10$ orders of magnitude.

# 2 Lecture 2: model-dependence of signals, classification of energy spectra

## 2.1 Alternative cosmological histories, velocity suppression and enhancement

In the estimates above, we have focused on the simplest cases of annihilation (rate scales as density squared) and decay (rate scales as density), using the thermal relic scenario as a convenient benchmark normalization in the annihilation case. However, in general DM models, the particle injection rate may have a different dependence on density, and may additionally depend on other environmental factors – most commonly, the velocity dispersion of the DM particles. This gives rise to a different redshift dependence, requiring us to redo these estimates.

### 2.1.1 Asymmetric DM and coannihilation

There are certainly "nightmare scenarios" for indirect detection where it will be quite difficult to see a signal in the present day. Perhaps the simplest of these is the possibility that DM is asymmetric, i.e. DM and anti-DM are distinct particles with different abundances (anti-DM is taken to be less abundant, without loss of generality). Then if the annihilation cross section is sufficiently large, the annihilation process will freeze out due to a lack of anti-DM before it would decouple in the symmetric case. This requires an annihilation cross section larger than the standard thermal relic value of $\langle \sigma v_{\text{rel}} \rangle \approx 2 \times 10^{-26}$ cm$^3$/s, but once this requirement is satisfied, the final DM abundance is set by the asymmetry rather than the annihilation cross section. This is analogous to how the relic abundance of ordinary matter is determined.

It is worth noting that the annihilation rate need only be slightly larger than the thermal relic value for this scenario to work, and for the anti-DM to be depleted to a level far below the relic density [12]. In the standard thermal freezeout scenario every DM particle is annihilating against a target whose density is also Boltzmann-suppressed, but this is not true for anti-DM in an asymmetric scenario, since the DM abundance converges toward its minimum value set by the asymmetry, rather than toward zero. Thus the remaining anti-DM has many available targets to annihilate on, and is efficiently depleted by even modest annihilation rates.

In the simplest models, the indirect detection signal at late times is very small, as the abundance of anti-DM is exponentially suppressed. However, if the anti-DM population can be regenerated at some later time (e.g. Refs. [13–15]), there can potentially be very large indirect-detection signals due to the large annihilation cross section. A comprehensive review of asymmetric DM is given in Ref. [16].

More generally, DM annihilation at early times can rely on a partner particle that is absent at late times. A widely-studied model of DM is the "vector portal", where the DM is a dark Dirac fermion charged under a dark U(1) gauge group. The dark U(1) gauge boson, called a "dark photon", has a non-zero mass, but mixes with the SM photon. Once the U(1) gauge symmetry is broken, it is possible (via a number of different models) to split the Dirac fermion (four degrees of freedom) into two nearly-degenerate Majorana fermions $\chi_1$ and $\chi_2$ (two degrees of freedom each). In this "pseudo-Dirac DM" scenario, because Majorana fermions do not carry conserved charge, the coupling between the dark photon and the dark fermions must involve one $\chi_1$ particle and one $\chi_2$ particle. If the dark photon is heavier than twice the DM mass, the dominant annihilation process into SM particles involves $\chi_1 + \chi_2$ annihilating into an off-shell dark photon which can convert to a SM fermion-antifermion pair via its mixing with the photon. However, at late times, the heavier of the two $\chi$ states (without loss of generality, call it $\chi_2$) can decay to the lighter state by emitting an off-shell dark photon. Once the vast majority of the $\chi_2$ states have decayed, this annihilation channel becomes strongly suppressed. Indirect detection constraints on such "pseudo-Dirac" models are thus significantly weakened – although there can still be non-negligible limits from other annihilation channels, as discussed in Ref. [17].

This is an example of a more general framework called "coannihilation", where at early times the DM may interact with, annihilate against or be partially comprised of another species, which no longer exists in the late universe (see e.g. Ref. [18] for a discussion). The presence of this partner species during freezeout modifies the annihilation rate. As another example, pure wino DM in supersymmetric models consists of a neutral Majorana fermion, but there is a slightly heavier chargino state present in the spectrum of the theory; during freezeout, both are present, as the wino-chargino mass splitting is small compared to the temperature at freezeout. Consequently the relic abundance of both the wino and chargino species needs to be computed (including annihilations that involve both winos and charginos simultaneously), and added together; at late times the chargino decays to the wino. The presence of such additional states in the early universe can either decrease or increase the late-time DM annihilation cross section relative to expectations from the simple thermal relic scenario, depending on the relative sizes of the various relevant annihilation rates.

### 2.1.2 Low-velocity suppression and small-coupling models

Another possibility is that annihilation from an initial state with total orbital angular momentum $L = 0$ ($s$-wave annihilation) is strongly suppressed, so that the dominant annihilation during freezeout occurs from $L \geq 1$ initial states. The contributions to annihilation from such states scale as $\sigma v_{\mathrm{rel}} \propto v_{\mathrm{rel}}^{2L}$, at leading order in velocity. For $p$-wave annihilation ($L = 1$), the most common example, $\sigma v_{\mathrm{rel}}$ is suppressed by $v_{\mathrm{rel}}^2 \sim T/m_{\mathrm{DM}}$, which is a factor of $\mathcal{O}(1/20)$ at freezeout as discussed above, but $\mathcal{O}(10^{-6})$ in the present-day Galactic halo, where the typical velocity of DM particles is $v \sim 10^{-3}c$.

A good summary of when the $s$-wave contribution is absent or suppressed is given in Ref. [19]. For one example, consider Majorana fermion DM annihilating to SM scalars (i.e. Higgs fields). Because the particles in the initial state are identical fermions, their overall wavefunction must be antisymmetric. If their total orbital angular momentum $L = 0$, the spatial part of the wavefunction is symmetric, so the spin configuration must be antisymmetric; i.e. they must be in the spin-singlet state with $S = 0$. By angular momentum conservation, the final state must have $J = 0$, and since the constituents are scalars with no spin, they must also have $L = 0$. It follows that the initial state is CP-odd and the final state is CP-even, so the $s$-wave contribution must involve CP violation; if the CP violation is small, this contribution will be suppressed.

Another classic example is the case of Majorana fermion DM annihilating through a $s$-

channel $Z$ boson to light SM fermions $\bar{f}f$; again the initial state must have $S = 0$ if it has $L = 0$. If these interactions do not violate CP, then the final state must also have $L = 0$ and hence $S = 0$ by angular momentum conservation. Since the outgoing particles have opposite momenta and $S = 0$ implies their spins point in opposite directions, their helicities must be equal. But the $Z$ boson only couples to left-handed fermions and right-handed antifermions, so in the limit where $m_f \to 0$, this amplitude must vanish. Consequently, the $s$-wave contribution to the cross section is suppressed by powers of $m_f/m_{\mathrm{DM}}$.

An even more extreme version of low-velocity suppression occurs in scenarios where the freezeout occurs through multi-body annihilation, involving three or more DM particles in the initial state, or channels that are (exponentially) kinematically suppressed at late times (e.g. [20–22]). A less severe suppression occurs when the DM annihilates to a particle that is nearly degenerate with it, due to the small final-state phase-space (e.g. [23]).

Finally, of course, the DM may never have had appreciable interactions with the SM, and its abundance may be completely independent of its SM interactions. Axions and axion-like particles generally fall into this category. Freeze-in DM [24] is an example where the abundance is determined by DM-SM interactions, but these interactions are very weak and the DM was never in thermal equilibrium with the SM; classic indirect detection searches are not very useful for freeze-in DM, yet still cosmological probes are capable of testing large regions of the parameter space for these models [25, 26].

### 2.1.3 Low-velocity enhancement

On the other hand, there are classes of scenarios that are very *optimistic* for indirect detection, where the observable signals are enhanced and can be significantly larger than our estimates above. A simple example of this case occurs when the DM has some long-range attractive self-interaction. When the potential energy between DM particles due to this interaction exceeds the kinetic energy of the DM particles (i.e. at low velocities), and so comes to dominate the Hamiltonian, the annihilation rate can be greatly enhanced. This effect is called "Sommerfeld enhancement" (e.g. Ref. [27]).

Specifically, suppose the DM particles propagate in a long-range potential, approximated by a Coulomb potential with coupling $\alpha_D$, then the $s$-wave annihilation rate is enhanced by a factor $2\pi\alpha_D/v_{\mathrm{rel}}$ for $v_{\mathrm{rel}} \ll \alpha_D$. Higher partial waves, with angular momentum quantum number $L$ for the initial state, are enhanced by a factor of order $(\alpha_D/v_{\mathrm{rel}})^{2L+1}$ [28]. Of course, the DM-DM interaction is likely not infinite range; the mass of the force carrier $m_A$ can be neglected when $m_A \lesssim m_{\mathrm{DM}}v_{\mathrm{rel}}$, but for $v_{\mathrm{rel}} \lesssim m_A/m_{\mathrm{DM}}$, the enhancement generally saturates. Note that in order for a substantial enhancement to occur, we require there to be a range of velocities such that $\alpha_D \gtrsim v_{\mathrm{rel}} \gtrsim m_A/m_{\mathrm{DM}}$, i.e. we must have $m_A \lesssim \alpha_D m_{\mathrm{DM}}$. This criterion can be understood as a requirement that the Bohr radius $1/(\alpha_D m_{\mathrm{DM}})$ must be small enough to fit within the range of the Yukawa potential, $1/m_A$.

At specific values of $m_A/(\alpha_D m_{\mathrm{DM}})$, resonances occur, and the $s$-wave enhancement can instead be enhanced by a factor proportional to $1/v^2$, down to the saturation velocity. If such Sommerfeld enhancements are present, the prospects for indirect detection of thermal relics can be greatly enhanced, especially in searches where the typical DM velocity is very low (e.g. limits from the cosmic dark ages, before the DM has formed into gravitationally bound structures).

The long-range potential that allows for a Sommerfeld enhancement can also support bound states. The criterion for bound states to exist in a Yukawa potential is similar to the criterion for an appreciable Sommerfeld enhancement, $m_A \lesssim \alpha_D m_{\mathrm{DM}}$. Unstable bound states can serve as an additional channel for annihilation, and like Sommerfeld enhancement, the rate is enhanced at low velocities. Both Sommerfeld enhancement and bound-state formation can also modify the freezeout process itself, and hence the DM masses and couplings that give

the correct relic density.

These effects are not restricted to models with new dark forces. The classic WIMP is the lightest neutralino in a supersymmetric model (see Prof. Jonathan Feng's lectures for more detail). Neutralinos interact through the weak interaction; if the DM mass exceeds $m_W/\alpha_W \sim 2.4$ TeV, we expect Sommerfeld enhancement and bound states to be potentially relevant. Indeed, these effects greatly enhance the detectability of wino DM and DM in higher representations of the SM SU(2) electroweak gauge group (such as the quintuplet); we will return to this point in a few lectures (see section 4.2.3, and note that Fig. 8 demonstrates the resonance structure described above).

Temperature-dependent resonance effects can also increase the DM annihilation rate during freezeout without a commensurate increase in the late universe, or vice versa.

## 2.2 Beyond energy transfer: observing particles from annihilation/decay

Now let us begin talking about direct observation of the particles produced by annihilation and decay, in addition to the indirect effects on cosmic history discussed in the first lecture. We will begin by studying signals where directional information is absent or irrelevant, so all we know is the energy spectrum of the signal, and possibly how it varies as a function of time/redshift. These kinds of signals include contributions to isotropic background radiation of various frequencies, as well as charged cosmic rays measured in the neighborhood of the Earth (whose directions are scrambled by magnetic fields).

Let us begin with a simple example. Suppose DM has been decaying or annihilating for the whole history of the cosmos. Let us approximate the DM and the particles it produces as perfectly homogeneous,[1] and let us ignore interactions of those particles after production, so their abundance is only diluted by the expansion of the universe (this is a good approximation for neutrinos, and for photons in some energy ranges injected at sufficiently late times). Suppose we are interested in a photon signal and the DM produces $N$ photons per decay (after any unstable SM particles have decayed away). What is the rate of photons impinging on a detector today from this source?

These particles are traveling at the speed of light, so if we set up a detector of area $A$, it will see photons at a rate of $cAn_\gamma$, where $n_\gamma$ is the ambient number density of photons (let us ignore geometric factors for the directionality of the photons for now). Let $n_{\mathrm{DM}}$ be the present-day DM number density, and the scale factor today be $a_0$. Then the number of photons produced per unit time per unit volume, at some earlier time when the scale factor was $a$, will be $\Delta_\gamma = N n_{\mathrm{DM}}(a/a_0)^{-3} f_{\mathrm{dec}} H dt$, where as previously $f_{\mathrm{dec}}$ is the fraction of DM particles annihilating in a Hubble time. The resulting contribution to the present-day ambient density of photons $n_\gamma$ needs to be diluted by the expansion of the universe, $dn_\gamma = \Delta_\gamma (a/a_0)^3$, so overall we can write:

$$n_\gamma = n_{\mathrm{DM}} N \int_0^{t_0} dt\, H f_{\mathrm{dec}}(t). \tag{2}$$

Now note $H = (1/a)da/dt = d\ln a/dt$, so we can write:

$$n_\gamma = n_{\mathrm{DM}} N \int d\ln a\, f_{\mathrm{dec}}(a). \tag{3}$$

Note that exactly the same arguments for annihilation would just replace $f_{\mathrm{dec}}(t)$ with $f_{\mathrm{ann}}(t)$; we use the notation $f_{\mathrm{dec,ann}}$ to indicate that one should insert the appropriate $f$ value depending on whether the DM is decaying or annihilating. So the photon rate at the detector will be:

---

[1]Note that this can be an acceptable approximation for decaying DM or annihilating DM at sufficiently early times, but it will generally badly underestimate the total annihilation rate for annihilating DM at late times, where most annihilation takes place in structures with higher density than the average.

$$R_\gamma = n_{\text{DM}} N c A \int d \ln a \, f_{\text{dec,ann}}(a) = \frac{\rho_{\text{DM}} N c A}{m_{\text{DM}}} \int d \ln a \, f_{\text{dec,ann}}(a). \tag{4}$$

The cosmological DM density is about $\rho \sim 10^{-6}$ GeV/cm$^3$, as we mentioned last time. Let's suppose $N = 10$, we have a 1 m$^2$ detector, and we're looking for 100 GeV DM. Then we find $R_\gamma \sim 3 \times 10^7 s^{-1} \int d \ln a \, f_{\text{dec,ann}}(a)$, or about $10^{15} \times \int d \ln a \, f_{\text{dec,ann}}(a)$ photons/year. So provided $f_{\text{dec,ann}}$ attains values around $10^{-15}$ or higher for an e-fold in the expansion history, we have the prospect of detecting at least a few events per year for 100 GeV DM (and the rate scales inversely with the DM mass).

This example is incomplete in a number of important ways. To name a few:

- This first-pass estimate assumes that injected particles do not interact. Particles will certainly lose energy via redshifting, and that can render them undetectable, as can other forms of energy loss. Interactions can produce significant numbers of secondary particles, which may be more or less detectable than their progenitor.

- The assumption of homogeneity means we have thrown out all information about the directionality of the events. In reality, regions of high DM density will have enhanced annihilation rates, as the rate will be determined by the average of the square of the DM density ($\langle \rho^2 \rangle$), not the square of the average of the DM density ($\langle \rho \rangle^2$). Directional information is also critical in separating signals from backgrounds.

- We have ignored all backgrounds: in reality, most channels will have significant backgrounds and detecting a handful of particles per year will not be sufficient.

We will address these issues in the rest of this lecture and the following lectures, but this initial example does indicate that we can hope to observe particles produced from even a very tiny fraction of annihilating and decaying DM, and that the resulting limits can potentially be stronger than the cosmological history bounds.

## 2.3 Spectra of annihilation/decay products

In the estimates from the last lecture, we made the (often very crude) approximation that all the energy liberated by DM annihilation or decay goes directly into our preferred search channel. This optimistic estimate is often not a good approximation (although it can provide a useful upper bound). The first step in working out the actual observable signal is usually to work out the spectrum of SM particles produced by annihilation, decay, or the other process of interest. Depending on the DM model, a wide range of SM particles can be produced; unstable SM particles decay on timescales that are prompt relative to almost all astrophysical/cosmological timescales, and so usually we can just consider the stable particles produced at the end of the decay chains.

Given a DM model, typically one first computes the rates for DM to annihilate or decay to the initial SM decay products, using standard perturbative methods. Extra theoretical ingredients may be needed at this stage for strongly-interacting DM or DM with long-range interactions, e.g. the Sommerfeld and bound-state corrections discussed last lecture. Then public codes such as Pythia and Herwig can be used to work out the eventual spectrum of stable SM particles that we observe. Results from these public codes have been tabulated, e.g. in the PPPC4DMID package ([29], http://www.marcocirelli.net/PPPC4DMID.html), including prescriptions for including radiative corrections and handling particle propagation (as we will discuss in the next section).

We often parameterize the possible particle spectra from annihilation/decay in terms of various possible 2-body SM final states (from the initial annihilation/decay), with the logic

that if annihilation/decay to 2-body final states *can* occur, then it will usually dominate the overall signal. This is not true in all cases – for example, the annihilation to two particles may be *p*-wave and hence velocity-suppressed, whereas adding a third particle to the final state lifts the velocity suppression and allows *s*-wave annihilation [30] – but it is common in many models. Then we can consider the spectra of particles produced by DM annihilation to pairs of quarks, gauge bosons, leptons etc, and their subsequent decays, and test these spectra against observations.

Very roughly speaking, there are four broad categories of SM final states:

1. Hadronic / photon-rich continuum: if the DM annihilates to $\tau$ leptons, gauge bosons, or any combinations of quarks, then copious neutral and charged pions will be produced in the subsequent decays of those particles. Neutral pions decay to a photon pair ($\pi^0 \to \gamma\gamma$) with a 99% branching ratio, so a broad spectrum of photons is produced. The charged pions decay dominantly to (anti)muons, neutrinos and antineutrinos; the (anti)muons decay to (anti)electrons, neutrinos and antineutrinos; thus these final states also imply copious production of neutrinos, electrons, and positrons with broad energy spectra. Given sufficient center-of-mass energy, heavier hadronic states can also be produced, including nuclei and anti-nuclei.

2. Leptonic / photon-poor: the DM annihilation produces mostly electrons and muons. Photons are produced directly only as part of 3-body final states, by final state radiation or internal bremsstrahlung; the rate for photon production is suppressed, and the photon spectrum is typically quite hard, peaked toward the DM mass [31]. (Note that similar hard photon spectra can be produced if the DM decays into a mediator that subsequently decays to photons, e.g. Ref. [32].) Copious charged leptons are produced, along with neutrinos in the case of the muon final state; these spectra also tend to be harder / more peaked than those produced by hadronic final states.

3. Photon lines: the DM annihilates directly to $\gamma\gamma$, or a monoenergetic photon plus another particle, or to another channel that produces a sharp peak in the photon signal as a function of energy. Such channels allow "bump hunts", and greatly reduce the possible astrophysical backgrounds; a clear detection of a gamma-ray spectral line would be very difficult to explain with conventional astrophysics. However, DM is known to carry no electric charge, and thus cannot couple directly to photons, so this signal must be suppressed at the 1-loop level at least, and is expected to be small. Any spectral peak with a width substantially smaller than the energy resolution of the relevant detector is observationally very similar to a line signal.

4. Neutrinos only: if the DM annihilates or decays solely to neutrinos, it is typically quite difficult to observe, although there are constraints from high-energy neutrino telescopes (e.g. [33]). DM annihilation within the Sun could lead to a signal that is neutrino-dominated because they are the only particles that escape (see e.g. Ref. [34] for a discussion of the signal modeling in this case). At sufficiently high DM masses, production of neutrinos implies that the electroweak gauge bosons can be radiated from the interaction point, and the decays of these gauge bosons give a contribution similar to that from hadronic states as discussed above. In some circumstances these radiative corrections are more detectable than the primary signal (e.g. [35]).

In Fig. 1 we show examples of the positron, photon, and electron neutrino spectra for a range of final states, reproduced from Ref. [29], to demonstrate these categories. These are the primary annihilation/decay spectra: i.e. they do not account for any propagation effects after the particles are produced. Note the relatively hard positron and photon spectra for

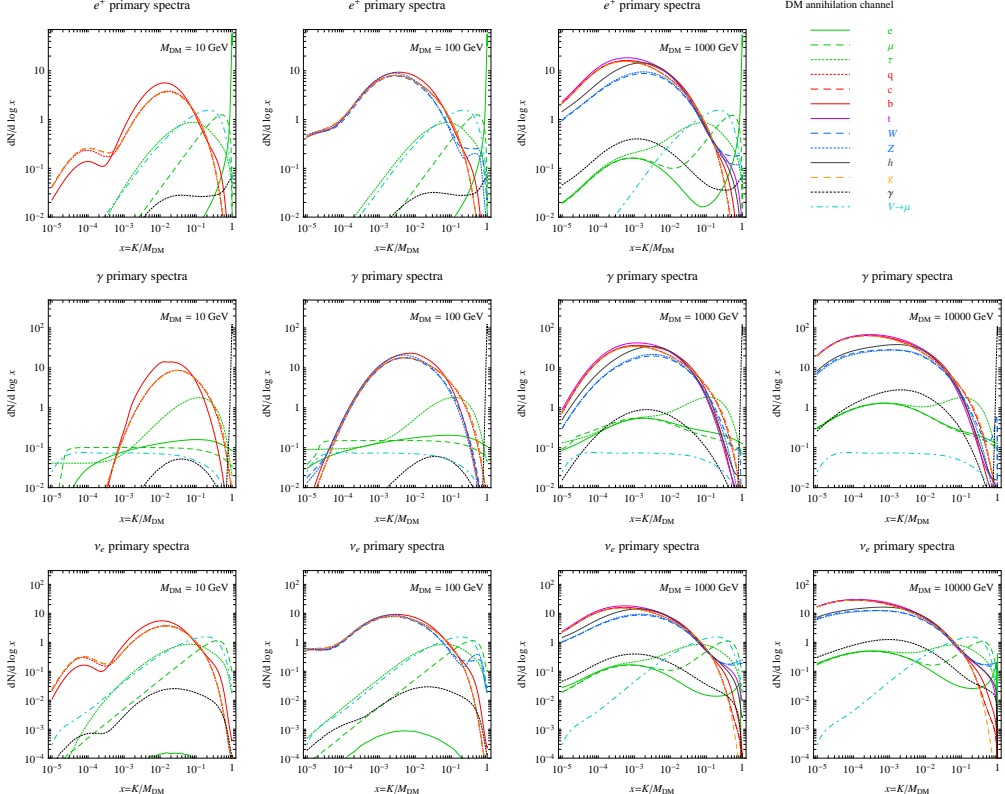

Figure 1: Primary fluxes of $e^+$ (top row), $\gamma$ (middle row) and $\nu_e$ (bottom row) from annihilation of DM with different masses, as a function of the particle (kinetic) energy $K$ divided by the DM mass. Different final states correspond to differently-colored lines as indicated in the legend. Reproduced from Ref. [29]; see that work for details.

annihilation to $e^+e^-$ in particular, and the hard neutrino spectrum produced by annihilation to leptonic states. Line signals are indicated by a sharply rising line where the particle energy approaches the DM mass, since delta functions cannot be plotted.

It is possible, of course, that the DM does not annihilate directly into SM particles; if the DM annihilates to other particles in a dark sector, and these subsequently decay back to the SM (either directly or through some cascade), then the eventual photon spectra need not lie in the space spanned by the 2-body SM final states. Some discussion of the range of possible spectra and the implications for indirect detection can be found in Ref. [36].

Charged particles from DM annihilation can also give rise to *secondary* photons, due to upscattering of ambient photons from starlight or the CMB, and synchrotron radiation from high-energy charged particles propagating in a magnetic field. For leptonic channels, this is often the largest photon signal; however, it depends on modeling the propagation of the charged particles, which we will discuss shortly.

## 2.4 Exercise 1: cosmology of Sommerfeld-enhanced DM annihilation

Suppose DM annihilates, and the annihilation rate develops a Sommerfeld enhancement of $\alpha_D/v_{\rm rel}$ where $v_{\rm rel}$ is the typical relative velocity between DM particles, for $v_{\rm rel} < \alpha_D \equiv 0.01$.

- Redo the estimates above and work out what fraction of DM annihilates per Hubble time as a function of redshift, in this case. You can first assume the DM fully decouples from the SM at freezeout and subsequently cools like a free non-relativistic particle. How

does the answer change if you instead assume the DM remains at the same temperature as the SM (both cooling with the expansion of the universe)?

- In these two cases, what DM mass scale would you expect to be able to exclude using CMB observations, using the sensitivity estimates above?

- For the first question, how would the answer change if instead the scaling of the Sommerfeld enhancement was $(\alpha_D/v_{\rm rel})^2$ for $v_{\rm rel} < \alpha_D$? Describe how the abundance of DM evolves in this case.

# 3 Lecture 3: particle propagation and J-factors

After calculating the spectra of particles produced by annihilation and decay, we still need to understand how those particles propagate to our telescopes, or produce secondary detectable signals. In some cases – for example when the particle is a photon or a neutrino, the origin point is relatively close (so the universe does not expand significantly over the travel time), and the particle's energy is such that the universe is effectively transparent, the observed spectrum may be essentially identical to the spectrum at production. However, this is generically untrue for charged particles, and depending on the signal, may not be true for photons and neutrinos either. Let us begin by studying two important example cases: the (1) propagation of charged particle cosmic rays in our galaxy, and (2) the absorption and secondary particle production due to injection of high-energy particles in the early universe.

## 3.1 Cosmic ray propagation

Charged particles produced by DM annihilation diffuse through the Galactic magnetic fields rather than following straight-line paths; furthermore, they can lose energy rapidly, so even on sub-Galactic scales, their spectrum changes with distance from the source. Both the signal and the background are thus theoretically challenging to model, and in the event of a possible signal being detected, there will be little spatial information as to the origin of the cosmic rays – their directionality is washed out by the ambient magnetic fields.

Public tools for modeling cosmic-ray propagation numerically include DRAGON [37, 38] and GALPROP [39, 40]. Here I will briefly sketch the principles of cosmic-ray diffusive propagation underlying these codes, following the review of Ref. [41]. Readers seeking a more detailed treatment may find it there.

Let us write the number density of cosmic rays at a given energy as $dn_{\rm CRs}/dE = \psi(\vec{x}, E, t)$. The evolution of this number density field is approximately governed by a diffusion equation:

$$\frac{\partial \psi}{\partial t} = D(E)\nabla^2 \psi + \frac{\partial}{\partial E}(b(E)\psi) + Q(\vec{x}, E, t). \tag{5}$$

Here $D(E)$ is a diffusion coefficient, which we approximate to be independent of position and time; $b(E)$ describes the energy losses for the cosmic-ray species in question; and $Q$ characterizes sources.

Other possible terms can be (and have been) added to this equation: convection of cosmic rays out of the Galactic plane can be described by a term of the form $\frac{\partial}{\partial z}(v_c \psi)$, where $v_c$ is the convection speed; decay or fragmentation of unstable cosmic rays can be described by a decay term $-\psi/\tau$; diffusive reacceleration corresponds to a term of the form $\frac{\partial}{\partial p}\left[p^2 D_{pp}\frac{\partial}{\partial p}\left(\psi/p^2\right)\right]$. But for the purposes of this lecture we will only consider the simple form of eq. 5.

The diffusion coefficient is generally parameterized as $D(E) = D_0(E/E_0)^\delta$, where $E_0 = 1$ GeV, $D_0 \sim$ few $\times 10^{28}$ cm$^2$/s. A number of different values for $\delta$ are in use, with $\delta \sim 0.3$ and

$\delta \sim 0.7$ being common bracketing values. $\delta = 1/3$ and $\delta = 1/2$ correspond to theoretical scenarios called Kolmogorov-type and Kraichnan-type diffusion, respectively, corresponding to different spectra for the magnetic field turbulence; higher values of $\delta \sim 0.7$ were preferred by earlier cosmic-ray data, e.g. Ref. [42], although the latest data seem to suggest a smaller value (e.g. Ref. [43] finds $\delta \sim 0.4 - 0.5$; Ref. [44] employs $\delta \sim 0.3 - 0.4$). In order to solve this equation, we also need to impose boundary conditions. A common approximation is to treat the Galaxy as a cylindrical slab with height $h$ (of order a few kpc) and radius $R$ (of order a few $\times$ 10 kpc), and impose a free-escape condition at the slab boundaries. The boundary parameters $R$ and $h$ and the diffusion parameters can be tuned to match measured cosmic-ray data.

At the level of dimensional analysis, the timescale for diffusion is characterized by $\tau_{\rm diff} \sim R^2/D(E)$, and the timescale for energy losses by $\tau_{\rm loss} \sim E/b(E)$. In this spirit, we can estimate $\partial/\partial E$ as yielding a factor of $-1/E$ – which is roughly appropriate for falling power-law spectra – and $\nabla^2 \psi$ as $-\psi/R^2$. If we assume a steady-state regime, then $\partial \psi/\partial t = 0$, and we can rewrite the diffusion equation (eq. 5) in the illustrative form:

$$-\frac{\psi}{\tau_{\rm diff}} - \frac{\psi}{\tau_{\rm loss}} + Q \approx 0. \tag{6}$$

This equation has the approximate solution $\psi \sim Q\min(\tau_{\rm diff}, \tau_{\rm loss})$.

There are two limiting cases, the diffusion-dominated regime where $\tau_{\rm diff} \ll \tau_{\rm loss}$, and the cooling-dominated regime where $\tau_{\rm loss} \ll \tau_{\rm diff}$. In the diffusion-dominated regime where energy losses are slow, which is the relevant case for protons and antiprotons, we have $\psi \propto Q(E)E^{-\delta}$, where $Q(E)$ describes the source spectrum of the cosmic rays. Thus diffusion softens the injected spectrum by an index set by $\delta$. The observed spectrum of protons has $dn/dE \propto E^{-2.7}$; if the injected spectrum for Galactic cosmic rays is $dn/dE \propto E^{-2}$, characteristic of particles accelerated by strong shocks [45], then this simple estimate would suggest $\delta \sim 0.7$.

In the cooling-dominated or loss-dominated regime, energy losses are fast relative to diffusion; this is the relevant regime for high-energy electrons and positrons. The main energy loss processes are synchrotron radiation in ambient magnetic fields, and inverse Compton scattering of ambient photons. If the cosmic rays are not too energetic – i.e. the geometric mean of their energy and the ambient photon energy is less than the electron mass – then the energy loss rate for inverse Compton scattering has a simple form, $dE/dt \propto E^2$; this is also the case for synchrotron radiation. Thus in this case we can write $b(E) \propto E^2$, and $\tau_{\rm loss} \propto E^{-1}$, vs $\tau_{\rm diff} \propto E^{-\delta}$. For $0 < \delta < 1$, as a consequence, losses increasingly dominate ($\tau_{\rm loss}$ is smaller) at higher energies. Thus we expect a spectrum of $Q(E)E^{-\delta}$ for low-energy electrons and positrons, breaking to $Q(E)E^{-1}$ at high energies.

As cosmic rays propagate through the Galaxy, protons can scatter on the ambient gas, producing secondary photons, positrons and electrons. For these secondary cosmic rays, $Q(E)$ should be replaced by the steady-state proton spectrum. If the original $Q(E) \propto E^{-2}$ for all species, and so the steady-state proton spectrum is $\propto E^{-(2+\delta)}$, then the secondary positron spectrum should have a spectrum of $E^{-(2+2\delta)}$ at low energies, breaking to $E^{-(3+\delta)}$ at high energies, in contrast to the primary positron spectrum, which is proportional to $E^{-(2+\delta)}$ at low energies and breaks to $E^{-3}$ at high energies. More generally, secondaries should have a softer spectrum than primaries by a factor of $E^{-\delta}$, if the primaries are in the diffusion-dominated regime.

Suppose that instead the source term is DM annihilation to $e^+e^-$, with the electron and positron each having energy equal to the DM mass: $Q(E) = Q_0 \delta(E - m_{\rm DM})$. In this case the approximation of the injected spectrum as a power law is clearly inaccurate. Let us consider

the steady-state spectrum in the loss-dominated regime, so the diffusion equation becomes:

$$\frac{\partial}{\partial E}(b(E)\psi) = -Q_0\delta(E - m_{\mathrm{DM}}).\tag{7}$$

Integrating both sides gives:

$$\psi(E) = \frac{Q_0}{b(E)}, \quad 0 \le E \le m_{\mathrm{DM}}.\tag{8}$$

So for $b(E) \propto E^2$ as discussed above, $\psi(E) \propto Q_0 E^{-2}, 0 \le E \le m_{\mathrm{DM}}$; that is, the steady-state spectrum is a smooth power law with a sharp cutoff at the DM mass. Due to the relatively hard spectrum – harder than one would expect from shock-accelerated cosmic rays softened by diffusion and/or losses – combined with the sharp endpoint, it may in principle be possible to distinguish such a signal from background. But this is quite non-trivial, as astrophysical sources can also have energy cutoffs, and experimental energy resolution is a limiting factor.

## 3.2 Particle cascades in the early universe

The first calculations of the ionization limits on DM annihilation and decay from CMB observations were performed in Refs. [46–48]. As we discussed last lecture, DM annihilation or decay during the cosmic dark ages can cause additional ionization of the ambient hydrogen gas; the resulting free electrons would scatter the CMB photons and modify the measured anisotropies of the CMB. In order to calculate this effect in detail, beyond the back-of-the-envelope estimates of last lecture, we need the following ingredients:

1. The spectrum of stable electromagnetically interacting particles produced by the DM annihilation/decay, and the redshift dependence of the energy injection.

2. A calculation of how these electromagnetically interacting particles cool and lose their energy, what fraction of their energy is converted into hydrogen ionization, and how long the cooling process takes.

3. A calculation of how extra ionizing energy modifies the ionization history of the universe, and how modifications to the ionization history affect the anisotropies of the CMB.

The third ingredient is available in public codes: RECFAST [49], HyREC [50] and CosmoRec [51] calculate the modified ionization history, while CAMB [52] and CLASS [53] can translate arbitrary ionization histories into modifications to the CMB anisotropy spectra. The first ingredient can usually be calculated fairly straightforwardly once the DM model is determined; it is the same spectrum-at-source relevant to other indirect searches. The second ingredient has been calculated and tabulated in Ref. [54] for electrons, positrons and photons, for keV-multi-TeV injection energies; Ref. [55] has performed a more limited calculation of the effect of protons and antiprotons and argues that their contribution to the ionization history will generally be small.

Note that the second ingredient here is agnostic as to the origin of the electromagnetically interacting particles, and the third ingredient does not require knowledge of the source of the extra ionization. Thus details of the particle physics model enter only in the first ingredient; separating the ingredients in this way thus allows the calculations in (2) and (3) to be worked out for arbitrary injections of electromagnetically interacting particles, and then applied to specific DM models as needed. The type of cascade calculated in (2) can also be relevant for considering high-energy particles injected into the present-day universe – for very high-mass DM decaying to SM particles, the cascade of lower-energy secondary particles can be more

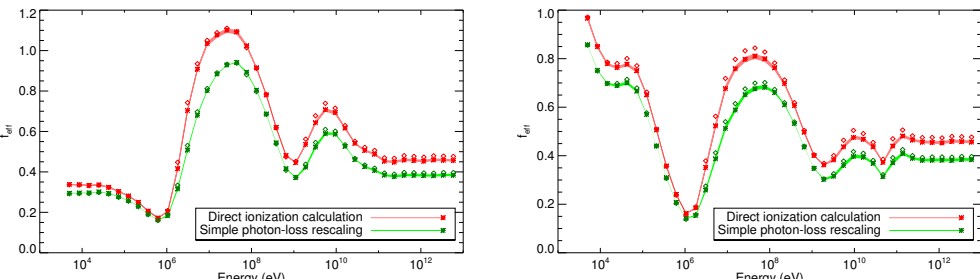

Figure 2: $f_{\text{eff}}$ coefficients as a function of energy for $e^+e^-$ (left column) and photons (right column), for annihilating DM. The widths of the red and green bands indicate systematic uncertainties in the derivation of the $f_{\text{eff}}$ factors. Red and green stars indicate two different methods of calculating the efficiency factors, with the red points corresponding to the detailed calculation and the green to a simplified version. Diamonds indicate $f_{\text{eff}}$ evaluated as the value of the efficiency of deposition at $z = 600$, rather than taking into account the full redshift dependence. Figure reproduced from Ref. [5].

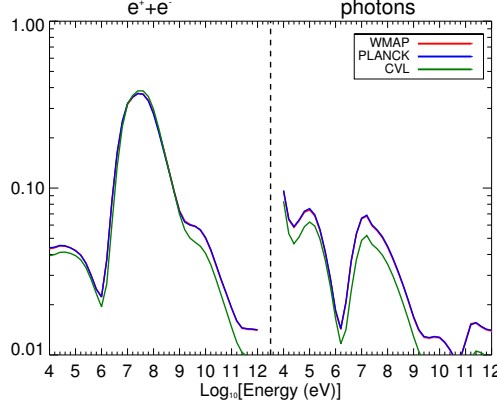

Figure 3: $f_{\text{eff}}$ coefficients as a function of energy for $e^+e^-$ (*left panel*) and photons (*right panel*), for decaying DM with a lifetime much longer than the age of the universe. Figure reproduced from Ref. [58].

detectable than the original particle, due to the large number of secondaries and the relative sensitivity of gamma-ray telescopes at different energies (see Ref. [56] for an example).

It turns out that the limit on $s$-wave annihilating DM from the CMB depends on essentially one number: the excess ionization at redshift $z \sim 600$ [4, 5, 57]. For decay, the signal is similarly dominated by redshift $z \sim 300$ [58, 59]. The shape of the CMB anisotropies is nearly model-independent; this parameter fixes the overall normalization. We can thus define an efficiency factor $f_{\text{eff}}$ such that the signal in the CMB is directly proportional to $f_{\text{eff}}\langle\sigma v_{\text{rel}}\rangle/m_{\text{DM}}$ for ($s$-wave-dominated) annihilation, or to $f_{\text{eff}}/\tau$ for decay, where $f_{\text{eff}}$ is a model-dependent efficiency factor. Recall that $\langle\sigma v_{\text{rel}}\rangle/m_{\text{DM}}$ ($1/\tau$) controls the rate of energy injection from annihilation (decay), as discussed earlier.

The parameter $f_{\text{eff}}$ depends primarily on how much of the injected power proceeds into electromagnetically interacting particles, as opposed to neutrinos. Secondarily, it depends on the spectrum of the injected electrons, positrons and photons; most of the variation occurs for particle energies below the GeV scale. Fig. 2 displays the numerically-computed $f_{\text{eff}}$ factors for photons and $e^+e^-$ pairs injected at different energies, for the case of $s$-wave annihilation [5]; Fig. 3 shows the equivalent factors for decay with a lifetime longer than the age of

the universe [58]. Results for arbitrary photon/electron/positron spectra can be obtained by integrating the product of the spectrum with $f_{\text{eff}}(E)$, to obtain an average $f_{\text{eff}}$ value. Note that the normalization in these figures is arbitrary; having set a constraint on any one reference DM model, one can convert the bound to a limit on any other DM model, by using the relative $f_{\text{eff}}$ values for the reference model and the model of interest.

For the case of annihilation, it is conventional to normalize $f_{\text{eff}}$ to the case of a reference model where 100% of the injected power is promptly absorbed by the gas, and roughly 1/3 of this power goes into ionization if the background ionization level is low [60]. Choosing $f_{\text{eff}} = 1$ for this reference model, CMB data can be used to set a limit on $p_{\text{ann}} \equiv f_{\text{eff}} \langle \sigma v_{\text{rel}} \rangle / m_{\text{DM}}$. The *Planck* Collaboration measured this limit as $p_{\text{ann}} < 3.2 \times 10^{-28} \text{ cm}^3 \text{ s}^{-1} \text{ GeV}^{-1}$ [61], updating a previous bound of $p_{\text{ann}} < 4.1 \times 10^{-28} \text{ cm}^3 \text{ s}^{-1} \text{ GeV}^{-1}$ [62]. Fig. 4 shows the implications of this earlier upper bound for $\langle \sigma v_{\text{rel}} \rangle$ as a function of $m_{\text{DM}}$, for various 2-body SM final states, using the curves shown in Fig. 2 to calculate the final-state-dependent and mass-dependent $f_{\text{eff}}$ factors. These limits can be updated to the 2018 *Planck* results [61] by a simple rescaling. Similar calculations can be performed for the case of decaying DM; see Ref. [58].

Because the CMB constraints measure total injected power, and the effect on the CMB anisotropy spectrum is essentially model-independent up to the overall normalization factor, these limits can be applied to a very wide range of DM models. In particular, they are often the strongest available constraints for DM masses and annihilation channels where the annihilation products are difficult to detect directly with current telescopes (e.g. because low-energy electrons and positrons are deflected by the solar wind, or low-energy photons are absorbed on their way to Earth, or we have no current telescopes observing the relevant energy range, or the astrophysical backgrounds are large and difficult to characterize).

One can also look at the modifications to the temperature history, instead of the ionization history. The required pipeline is quite similar: codes like RECFAST [49], HyREC [50] and CosmoRec can also calculate the modified temperature history in the presence of additional heating, and the amount of energy converted into heat due to the secondary particle cascade (given a spectrum of injected particles) has been tabulated and made public in Ref. [54]. The modified temperature history can then be compared directly to limits on the temperature of the universe from the Lyman-$\alpha$ forest or observations of primordial 21cm radiation, as discussed last lecture. There is one claim of such an observation [67], and many experiments seeking to measure the signal.

The CMB signal from annihilation or decay is dominated by relatively high redshifts of several hundred, but the potentially-observable changes to the temperature history depend primarily on physics at much lower redshifts, close to the redshift of measurement ($z \sim 3-20$). These signals are consequently much more sensitive to uncertainties in the physics of structure formation – which determines the average rate of annihilation in DM haloes during this epoch – and the ionization history during the epoch of reionization. In particular, changes to the ionization level of the gas can modify the cascade of secondary particles produced from the initial energy injection, so earlier energy injections can change the observability of later ones. These *backreaction* effects are captured in the DarkHistory code package [68], which tabulates the secondary particle cascade in an alternative way, and so allows it to be recalculated self-consistently with any modifications to the ionization and temperature history. These effects can be quite important when considering temperature changes due to energy injection signals near the current sensitivity limit. Current constraints from Lyman-$\alpha$ observations and forecast sensitivity for future 21cm observations can be found in Refs. [59, 69–71].

## 3.3 Directional information

Now let us consider the case where propagation/interaction effects are unimportant, and consequently particles travel on geodesics from their points of origin to our telescopes, allowing

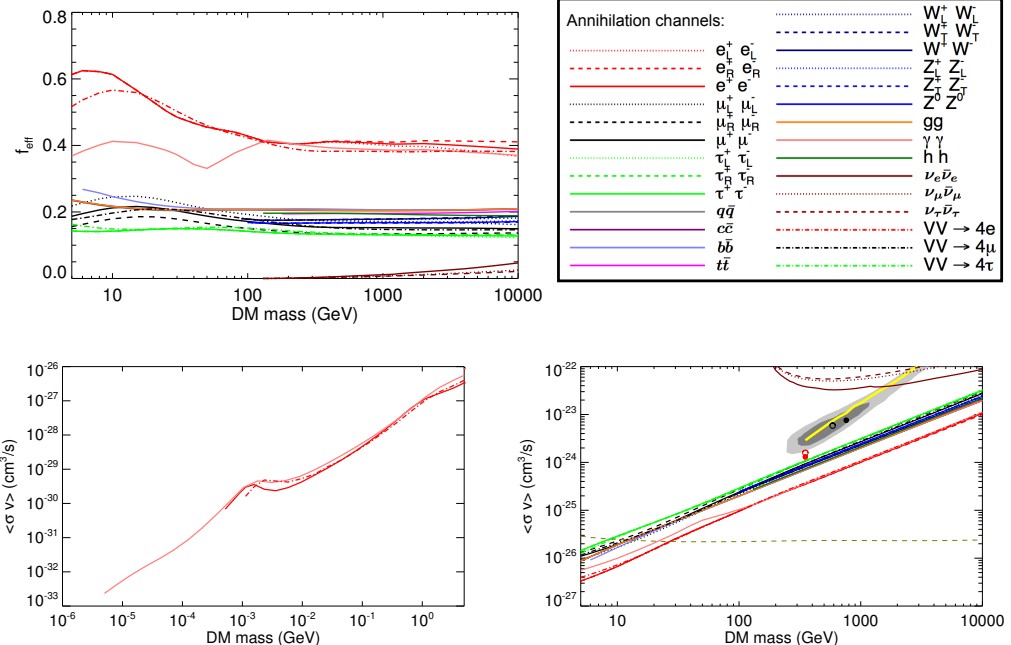

Figure 4: The *upper panel* shows the $f_{\text{eff}}$ coefficients as a function of DM mass for each of a range of SM final states, as indicated in the legend. The $VV \to 4X$ states correspond to DM annihilating to a pair of new neutral vector bosons $V$, which each subsequently decay into $e^+ e^-$, $\mu^+ \mu^-$ or $\tau^+ \tau^-$ (labeled by $X$). The *lower panels* show the resulting estimated constraints from recent *Planck* results [62], as a function of DM mass, for each of the channels. The *left panel* covers the range from keV-scale masses up to 5 GeV, and only contains results for the $e^+ e^-$, $\gamma\gamma$ and $VV \to 4e$ channels; the *right panel* covers the range from 5 GeV up to 10 TeV, and covers all channels provided in the PPPC4DMID package [29]. The light and dark gray regions in the lower right panel correspond to the $5\sigma$ and $3\sigma$ regions in which the observed positron fraction can be explained by DM annihilation to $\mu^+ \mu^-$, for a cored DM density profile (necessary to evade $\gamma$-ray constraints), taken from Ref. [63]. The solid yellow line corresponds to the preferred cross section for the best fit 4-lepton final states identified by Ref. [64], who argued that models in this category can still explain the positron fraction without conflicts with non-observation in other channels. The red and black circles correspond to models with $4e$ (red) and $4\mu$ (black) final states, fitted to the positron fraction in Ref. [65]; as in that work, filled and open circles correspond to different cosmic-ray propagation models. The near-horizontal dashed gold line corresponds to the thermal relic annihilation cross section [66]. Reproduced from Ref. [5].

us to identify the two-dimensional position of their point of origin on the sky. This is usually the case for neutrino signals, and sometimes for photons, depending on their energy and the distance to the point of origin. In this case we need to be able to predict the spatial distribution on the sky of the signal, not just its energy spectrum.

In general we have only a two-dimensional view of the sky, although in some circumstances we can discern the distance at which a particular photon was emitted, e.g. because we have redshift information. Thus what we observe, in general, will be the number of photons or neutrinos arriving at our detector from within a particular solid angle on the sky, within a particular time interval.

As previously, let us suppose our telescope/detector has area $A$, and consider the signal arising from a volume $dV$ located at coordinates $(r, \theta, \phi)$, where the Earth is at $r = 0$. Suppose each annihilation/decay produces an energy spectrum of photons (or neutrinos) $\left(\frac{dN_\gamma}{dE}\right)_0$. If the energy of the photons/neutrinos does not change between production and reception (i.e. redshifting, absorption etc are negligible), then the spectrum of photons received at Earth per volume per time is given by:

$$\frac{dN_\gamma}{dEdtdV} = \left(\frac{dN_\gamma}{dE}\right)_0 \frac{A}{4\pi r^2} \times \begin{cases} \frac{1}{2}\langle\sigma v_{\text{rel}}\rangle n(\vec{r})^2 & \text{annihilation} \\ \frac{n(\vec{r})}{\tau} & \text{decay} \end{cases}. \tag{9}$$

Integrating along the line of sight, we find:

$$\frac{dN_\gamma}{dEdtd\Omega} = \frac{A}{4\pi}\left(\frac{dN_\gamma}{dE}\right)_0 \times \begin{cases} \frac{\langle\sigma v_{\text{rel}}\rangle}{2m_{\text{DM}}^2}\int_0^\infty \rho(\vec{r})^2 dr & \text{annihilation} \\ \frac{1}{m_{\text{DM}}\tau}\int_0^\infty dr \rho(\vec{r}) & \text{decay} \end{cases}. \tag{10}$$

If the source of annihilation/decay is localized, it often makes sense to integrate over the solid angle subtended by the object, to obtain the full signal from that source:

$$\frac{dN_\gamma}{dEdt} = \frac{A}{4\pi}\left(\frac{dN_\gamma}{dE}\right)_0 \times \begin{cases} \frac{\langle\sigma v_{\text{rel}}\rangle}{2m_{\text{DM}}^2}\int dr d\Omega \rho(\vec{r})^2 & \text{annihilation} \\ \frac{1}{m_{\text{DM}}\tau}\int_0^\infty dr d\Omega \rho(\vec{r}) & \text{decay} \end{cases}. \tag{11}$$

Typically we separate the piece of this expression dependent on the particle physics from that entirely determined by the distribution of the DM mass density $\rho(\vec{r})$, which can be predicted from N-body simulations and/or measured by gravitational probes. The latter is called the "J-factor" of the source, and for annihilation can be defined as (note that there is more than one convention for the normalization in common use):

$$J_{\text{ann}} \equiv \int dr d\Omega \rho(\vec{r})^2, \tag{12}$$

so that $\frac{1}{A}\frac{dN_\gamma}{dEdt} = \frac{1}{8\pi}\frac{\langle\sigma v_{\text{rel}}\rangle}{m_{\text{DM}}^2}\left(\frac{dN_\gamma}{dE}\right)_0 J_{\text{ann}}$.

There is an additional simplification that can be applied if the source is spherically symmetric. If $d$ is the distance from the Earth to the center of the source, $r$ is (as previously) the distance from the Earth to the point of annihilation, and we choose the $z$-axis of our coordinate system to point in the direction of the center of the source, then spherical symmetry of the source implies that $\rho(\vec{r})$ is in fact $\rho(\sqrt{r^2 + d^2 - 2dr\cos\theta})$. The integral over $d\phi$ can then be performed immediately. If the source is the center of the Galaxy and we are working in Galactic coordinates $l$ and $b$ ($l$ is the Galactic longitude defined so the Galactic center is at $l = 0$, $b$ is the latitude expressed as the angle from the equator), then it is helpful to note that $\cos\theta = \cos l \cos b$.

The J-factors of different sources characterize the relative size of their expected annihilation signals. Especially for regions close to the centers of halos, the J-factor can depend sensitively on the presumed density profile; a common choice is to model halos as following the Navarro-Frenk-White (NFW) profile [72], $\rho \propto r^{-1}/(1 + r/r_s)^2$, where now $r$ denotes distance from the center of the halo and $r_s$ is a characteristic scale radius. Under this assumption, the dwarf satellite galaxies of the Milky Way have J-factors in the neighborhood $J_{\text{ann}} \approx 10^{17-20}$ GeV$^2$/cm$^5$ [73]; the region within 1 degree of the Milky Way's center has $J_{\text{ann}} \approx 10^{22}$ GeV$^2$/cm$^5$.

Other density profiles that are commonly used are the Einasto profile [74], with density given by $\rho(r) = \rho_0 e^{-\left(\frac{2}{\alpha}\right)\left(\left(\frac{r}{r_{-2}}\right)^\alpha - 1\right)}$ and the Burkert profile [75] $\rho(r) = \frac{\rho_0}{(1+r/r_s)(1+r^2/r_s^2)}$. The latter can be used to describe profiles with a flat-density core for $r < r_s$.

Naively we would thus expect the Galactic Center to be a more promising target for annihilation searches than dwarf galaxies. However, astrophysical backgrounds in the Galactic Center and the surrounding region are also much higher than in dwarf galaxies, so it can be more difficult to distinguish any potential signal from the background. Dwarf galaxies contain few baryons, so are relatively clean targets for indirect searches. The typical velocity of DM particles in dwarfs is also much smaller than in the Galactic Center; this can reduce signals in dwarfs in models where the annihilation is suppressed at low velocities (e.g. where $p$-wave annihilation dominates), or enhance it in models where the converse is true (e.g. models with Sommerfeld enhancement). The expected signal at the Galactic Center also depends strongly on the assumed model for the Milky Way's DM density profile; however, if a possible signal were observed, it would be possible to infer information about the DM density profile from the morphology of the signal.

The J-factors listed above include only the contribution from the smooth NFW density profile; in reality, the presence of small-scale substructure could potentially greatly increase $J_{\mathrm{ann}}$. DM halos are thought to form by accretion of many smaller halos that formed at earlier times, and annihilation can be further enhanced in these small dense structures (because $\langle \rho^2 \rangle \neq \langle \rho \rangle^2$, and the former is the relevant quantity for annihilation). The effect grows with the size of the host halo (as larger halos can contain more substructure), and for galaxy clusters could potentially give rise to a $\mathcal{O}(10^3)$ enhancement to $J_{\mathrm{ann}}$ [76,77] (although more recent studies suggest a smaller enhancement [78]). However, the size of this "boost factor" is highly uncertain, as models predicting large enhancements tend to have most of the annihilation power arising from subhalos well below the mass scale which can resolved in simulations, i.e. $10^{5-6}$ solar masses.

For the case of decay, substructure is irrelevant, and the signal size is controlled by $\int \rho(\vec{r}) dr\, d\Omega$. If the source is distant, so the distance from Earth to every point in the source is approximately equal at $r \approx R$, then this integral becomes approximately $\frac{1}{R^2} \int \rho(\vec{r}) dV = M/R^2$, where $M$ is the total mass of the source. Thus the strongest signals come from targets that have large total DM mass and are also relatively close; some of the strongest constraints arise from study of galaxy clusters.

So far we have assumed that the spectrum of neutrinos/photons produced by DM annihilation, followed by prompt decays of the annihilation products, propagates to Earth essentially un-distorted. This is a good approximation for neutrinos and gamma-rays from our own Galaxy and nearby systems. However, for more distant targets, we must consider redshifting, and possibly absorption.

For a simple example, let us return to the case of the isotropic signal from annihilation of DM in the intergalactic medium, where we make the simplifying assumption that the density is equal to the overall cosmological DM density (note this will significantly underestimate the average all-sky contribution to annihilation from late redshifts, where much annihilation takes place in high-density bound structures). Let us now take into account the evolution of the spectrum with redshift rather than simply counting the number of photons. Now we must integrate over photons (or neutrinos) originating from all possible redshifts; we are interested in the photon density and spectrum in a present-day volume $dV_0$, arising from annihilation at all earlier times. We can write:

$$\frac{dN}{dE\, dV_0} = \int_{\infty}^{0} dz\, \frac{dt}{dz} \frac{dN_{\gamma}(z)}{dE} \frac{\langle \sigma v_{\mathrm{rel}} \rangle}{2} n(z)^2 \frac{dV_z}{dV_0} \,. \tag{13}$$

Here $dV_z$ is the physical volume at redshift $z$ corresponding to the same comoving volume as $dV_0$. Since $a = 1/(1+z)$, $dV_z/dV_0 = (a/a_0)^3 = 1/(1+z)^3$. We are exploiting the fact that the photon number density is (in this case) the same everywhere in the universe, and the photon number per comoving volume is preserved under the cosmic expansion, to argue that

the average photon number density originating from DM annihilation at redshift $z$ is depleted exactly by $(1+z)^3$ by the present day. Note also that $\frac{dN_\gamma(z)}{dE}$ on the right-hand side is the spectrum of photons produced by an annihilation at redshift $z$ which have energy $E$ *today*. If we define $E_z = E(1+z)$, i.e. the energy of a photon at redshift $z$ if that photon has energy $E$ today, then $\frac{dN_\gamma(z)}{dE} = (1+z)\frac{dN_\gamma(z)}{dE_z} = (1+z)(dN_\gamma/dE')_0|_{E'=E_z}$. The factor of $dt/dz$ is needed to convert the rate of annihilations per unit time into annihilations per change in redshift; note that $d/dt \ln(1+z) = -d/dt \ln a = -H(z)$, so $dt/dz = -1/(H(z)(1+z))$. Finally, the cosmological DM density $n(z)$ scales as $a^{-3}$, i.e. $(1+z)^3$.

Putting this all together, we obtain:

$$\frac{dN}{dEdV_0} = \int_0^\infty dz \frac{(1+z)^3}{H(z)} \rho(z=0)^2 \left[ \left( \frac{dN_\gamma}{dE'} \right)_0 \bigg|_{E'=E(1+z)} \frac{\langle \sigma v_{\rm rel} \rangle}{2m_{\rm DM}^2} \right]. \tag{14}$$

The term in square brackets encapsulates the model-dependent particle physics. For decay, we would replace $\langle \sigma v_{\rm rel} \rangle/2m_{\rm DM}^2$ with $1/m_{\rm DM}\tau$, $\rho(z=0)^2$ with $\rho(z=0)$, and the $(1+z)^3$ factor with $(1+z)^0$; the result is otherwise the same.

As an example of using this result, consider annihilation to a pair of photons with energies equal to the DM mass, so $(dN/dE)_0 = 2\delta(E - m_{\rm DM})$. Then we have:

$$
\begin{aligned}
\frac{dN}{dEdV_0} &= \frac{\langle \sigma v_{\rm rel} \rangle}{2m_{\rm DM}^2} \rho(z=0)^2 \int_0^\infty dz \frac{(1+z)^3}{H(z)} 2\delta(E(1+z) - m_{\rm DM}) \\
&= \frac{\langle \sigma v_{\rm rel} \rangle}{2m_{\rm DM}^2} \rho(z=0)^2 \frac{(m_{\rm DM}/E)^3}{H(z=m_{\rm DM}/E-1)} \frac{2}{E}, \quad 0 \le E \le m_{\rm DM} \\
&= m_{\rm DM} \langle \sigma v_{\rm rel} \rangle \rho(z=0)^2 \frac{1}{H_0 \sqrt{\Omega_m (m_{\rm DM}/E)^3 + \Omega_\Lambda}} \frac{1}{E^4}, \quad 0 \le E \le m_{\rm DM},
\end{aligned}
\tag{15}
$$

where in the last line we have neglected the contribution to $H$ from the radiation field, which is valid for small $z$.

In the general case, we will need to include both non-uniformity and redshifting, obtaining:

$$\frac{dN_\gamma}{dEdAdt} = \int \frac{d\Omega}{4\pi} \int dz \left( \frac{dN_\gamma}{dE'} \right)_0 \bigg|_{E'=E(1+z)} \frac{1}{H(z)(1+z)^3} \times \begin{cases} \frac{\langle \sigma v_{\rm rel} \rangle}{2m_{\rm DM}^2} \rho(z,\theta,\phi)^2 & \text{annihilation} \\ \frac{1}{m_{\rm DM}\tau} \rho(z,\theta,\phi) & \text{decay} \end{cases}. \tag{16}$$

The special cases discussed above can be obtained by taking $\rho(z,\theta,\phi) = \rho(z=0)(1+z)^3$ on one hand (for the isotropic homogeneous case), or by setting $z=0$ to neglect redshifting and noting that:

$$dz/H(z) = (1+z)d\ln(1+z)/H(z) = -(1+z)dt = -dt, \tag{17}$$

and then replacing $-dt$ with $dr$ for particles traveling toward Earth at lightspeed.

To include absorption, we could include a factor of the form $e^{-\tau(E,z)}$ inside the integral, where the function $\tau(E,z)$ describes the optical depth for a photon emitted at redshift $z$ and with (measured at $z=0$) energy $E$. Non-uniform sources, redshifting and absorption can all be relevant when computing contributions to the ambient radiation fields from DM annihilation/decay over the history of the universe; see e.g. Ref. [79–81] for examples.

### 3.4 Exercise 2: CMB limits on arbitrary DM models

Consider a model of DM that annihilates with equal branching ratios into tau leptons, muons, and electrons.

- Use http://www.marcocirelli.net/PPPC4DMID.html, or else your favorite event generator, to predict the spectrum of photons, electrons and positrons produced by this model at the point of annihilation, for a DM mass of 100 GeV. Try to write your code so it is easily generalizable to other DM masses. Plot the spectra.

- If you have access to Mathematica, download the notebook at

  https://faun.rc.fas.harvard.edu/epsilon/detaileddeposition/annihilation/

  Use it to estimate the efficiency factor $f_{\text{eff}}$ relevant for CMB constraints, for your model as a function of the DM mass. (If you do not have Mathematica, you can use the .dat files at the same link; please contact the author of these notes at tslatyer@mit.edu if you need help on how to proceed.) You can use the "3 keV" prescription for normalizing $f_{\text{eff}}$, since this is what the *Planck* collaboration used in setting their bounds.

- Assuming a thermal relic annihilation cross section, what mass range can you exclude using the latest bounds from *Planck*, which can be found in Section 7 of [61]?

# 4 Lecture 4: considerations for current indirect searches

## 4.1 Some comments on backgrounds

Astrophysical backgrounds for signals from DM vary depending on the particle species and energy, and hence on the DM mass. For sufficiently high-energy (gamma-ray) lines or sharply-peaked spectra, backgrounds are essentially non-existent; the only challenge is collecting sufficient statistics. For continuum gamma rays, cosmic rays interacting with the gas and starlight produce background photons – hadron-hadron collisions produce neutral pions which decay to gamma rays, and cosmic rays upscatter ambient photons to gamma-ray energies. Pulsars also produce photons with energies of a few GeV and below. At X-ray energies, relevant for searches for sterile neutrino DM, there are continuum X-rays from hot gas, as well as spectral lines from various atomic processes. At radio and microwave energies, relevant for searches for synchrotron radiation from weak-scale DM annihilation products, backgrounds include the CMB, synchrotron radiation from conventional sources, and thermal emission from interstellar dust.

For charged particles, a typical search strategy is to look for antimatter rather than the corresponding matter species, since the antimatter background is much lower. Nonetheless, positrons and antiprotons are regularly produced through cosmic-ray collisions in the Galaxy, and this provides a non-negligible astrophysical background for positrons and antiprotons from DM. Antideuterons and heavier nuclei are expected to have essentially zero background from SM processes, but are also expected to be much rarer products of DM annihilation and decay. There has been considerable discussion recently about uncertainties in the rate for production of antinuclei from DM annihilation/decay, prompted in part by a tentative claim by the AMS-02 experiment to have observed several antihelium nuclei – we will address this again later on.

## 4.2 A selection of indirect searches for dark matter

In the previous lectures we have argued that DM annihilation (decay) at interesting cross sections (lifetimes) can have observable traces in the present day and over the history of the universe. We have described analytic and numerical tools you can use to translate from DM models into observable cosmological and astrophysical signals. Let us now summarize some

of the leading constraints obtained by applying those methods, in addition to the cosmological limits discussed earlier in these notes.

Fig. 5 summarizes the energy reach of a number of current and near-future telescopes, several of which will be discussed below, spanning photon energies from radio to gamma-rays, as well as neutrino and cosmic-ray detectors. These plots are reproduced from Ref. [82].

### 4.2.1 WIMP annihilation limits from gamma rays

Observations of the Milky Way's dwarf galaxies by *Fermi* [83] provide some of the most robust and stringent bounds on weak-scale DM annihilating to photon-rich channels. These limits are publicly available as likelihood functions for the flux in each energy bin, allowing constraints to be set on arbitrary spectra;[2] for example, for annihilation to *b* quarks, the thermal relic cross section is constrained for DM masses below ∼ 100 GeV.

The dwarf galaxies have very low baryonic matter content, meaning the expected astrophysical backgrounds associated with the dwarfs themselves are very small (see e.g. Ref. [84] for a study of backgrounds associated with pulsars in dwarf galaxies). However, observing the dwarf galaxies still requires looking through the Milky Way's halo of diffuse gamma rays, so there are non-zero astrophysical backgrounds. Observations of dwarf galaxies in gamma rays typically involve an integration over much of the volume of the dwarf, and so are not especially sensitive to the details of the density profile in the innermost regions; however, there can still be large uncertainties in the J-factors due to uncertainties in the DM content and distribution (e.g. Ref. [73]). Using data-driven estimates for the J-factor uncertainties and astrophysical backgrounds, Ref. [85] recently found that the constraints can weaken by a factor of a few compared to earlier calculations (for example, for annihilation to *b* quarks they find the thermal relic cross section is only excluded for DM masses below ∼ 30 GeV).

Limits of similar strength, but with different (and potentially more severe) systematic uncertainties, can be obtained from a range of other gamma-ray searches with *Fermi* data, including observations of the Milky Way halo (e.g. [86]), of galaxy groups (e.g. [87]), and of the extragalactic background radiation (e.g. [88]).

The VERITAS, MAGIC, HAWC and H.E.S.S telescopes also set constraints on these channels from similar dwarf studies [89–92], which come to dominate those from *Fermi* for DM masses well above 1 TeV. Stronger high-energy limits have been presented using data from the H.E.S.S telescope [93, 94], although these rely on studies of the region around the Galactic Center, and are thus more sensitive to uncertainties in the DM density profile.

Note that as a general rule, space-based telescopes are needed for high sensitivity to gamma rays below $\mathcal{O}(100)$ GeV: in this energy window, *Fermi* plays a crucial role, and the fact that it is a full-sky telescope allows blind searches for signals and studies of large-scale diffuse emission. Above this scale, air Cherenkov telescopes (ACTs), such as H.E.S.S, VERITAS and MAGIC, typically set the strongest constraints: these telescopes are ground-based and so can have much larger effective areas than space-based telescopes, although they typically have small fields of view so need to perform targeted observations. At even higher energies, water Cherenkov telescopes such as HAWC and LHAASO can take over: they have higher energy thresholds, but large fields of view, allowing for long exposures on a wide range of targets.

### 4.2.2 WIMP annihilation limits from cosmic rays

The AMS-02 instrument has presented measurements of the spectrum of a wide range of cosmic ray species, at the location of the Earth. For DM limits the most relevant channels are positrons [95] and antiprotons [96–98], although measurements of other cosmic rays help constrain the propagation parameters discussed in previous lectures. For example, Ref. [99]

---

[2]https://www-glast.stanford.edu/pub_data/

Figure 5: Summary of selected experiments by year operating, and approximate energy sensitivity. All figures are reproduced from Ref. [82]; see that work for further detail.

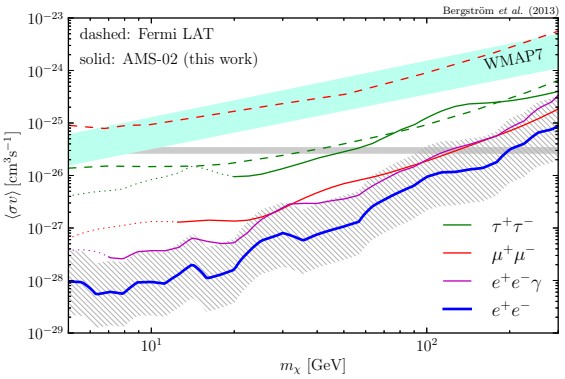
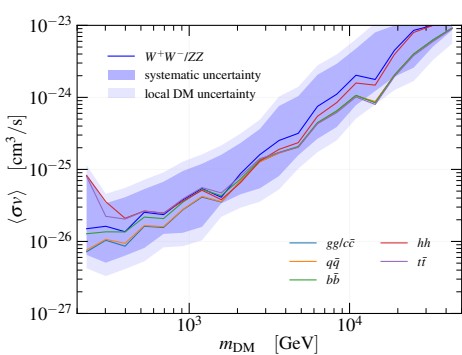

Figure 6: Limits from AMS-02 data on DM annihilation to (left) leptonic channels and (right) hadronic channels. In the *left panel*, dashed lines indicate bounds from *Fermi* observations of dwarf galaxies; the dotted portions of the solid constraint curves from AMS-02 are potentially affected by solar modulation effects. The hatched band around the $e^+e^-$ constraint line indicates the estimated uncertainty due to systematic uncertainties in the local DM density and energy loss rate. In the *right panel*, the different colored lines represent the nominal constraint from antiproton observations for different hadronic final states, and the blue bands denote systematic uncertainties in the limit for annihilation to W and Z bosons. Reproduced from Ref. [95] (*left panel*) and Ref. [96] (*right panel*).

uses beryllium and boron measurements to constrain the galactic halo size, and Ref. [100] provides new benchmark parameters for DM searches.

Fig. 6 displays limits on DM annihilation from AMS-02 measurements of positrons and antiprotons, which provide sensitive probes – potentially more sensitive than the dwarf searches – of leptonic and hadronic annihilation channels respectively. These constraints are subject to substantial systematic uncertainties, associated with cosmic-ray propagation, the effects of the Sun's magnetic field, and (in the hadronic case) the production cross section for antiprotons. However, current estimates of those systematic uncertainties still allow antiproton observations to test the thermal relic cross section for annihilating DM up to several hundred GeV in mass (with a gap in sensitivity around 40-130 GeV, corresponding to an excess which we will discuss later).

There are also limits on WIMP annihilation from radio observations; the signal mechanism involves the production of electrons and positrons, which produce synchrotron radiation (typically in the radio or microwave bands) in ambient magnetic fields. These constraints inherit uncertainties on the cosmic-ray propagation, and also depend sensitively on the magnetic field. However, they can potentially be very stringent, probing the thermal relic cross section for DM masses up to 500 GeV (e.g. [101, 102]).

### 4.2.3 Line limits from the Galactic Center

For gamma-ray lines, as discussed above, astrophysical backgrounds are low. Thus the imperative is to optimize statistics, and it makes sense to look toward the Galactic Center. H.E.S.S [94, 103] and *Fermi* [104, 105] have presented limits on the possible gamma-ray line strength, as summarized in Fig. 7.

Note that while the usual expectation is that the line cross section will be well below the thermal relic value, there are caveats to this statement; in particular, if there are charged particles in the spectrum of the theory, close in mass to the DM, then the line cross section can be unexpectedly large. This is particularly true in cases where a long-range potential

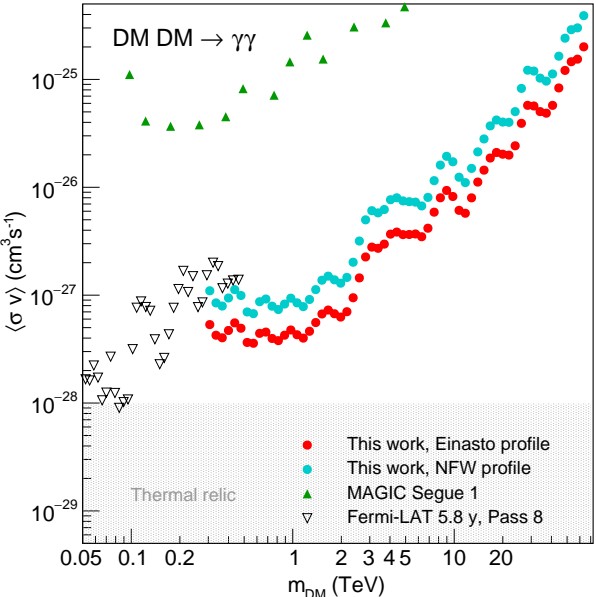

Figure 7: Upper bounds on the cross section for DM annihilation to $\gamma\gamma$, from *Fermi* (black triangles), MAGIC (green triangles), and H.E.S.S (red and cyan dots). Red and cyan dots correspond to the Einasto and NFW profiles respectively. The gray-shaded area is a theoretical forecast for the natural scale of the line cross-section. Figure reproduced from Ref. [94]; see that work for details.

couples two-particle DM states to two-particle states involving the charged particles – which is the case, for example, for pure wino DM in supersymmetric models. When $m_{DM} > m_W/\alpha_W$, the exchange of weak gauge bosons becomes effectively a long-range force, associated with a Sommerfeld enhancement that can readily be 1-2 orders of magnitude; the line cross section can be enhanced even further, since the long-range W-exchange potential allows any pair of winos to effectively oscillate into a chargino-chargino state, which can annihilate to $\gamma\gamma$ at tree level (see Fig. 9).

Fig. 8 shows how the prediction for the line cross section for wino DM compares to current and future constraints, assuming the pure wino constitutes 100% of the DM (this requires non-thermal production for masses not in the 2-3 TeV range). Even if the DM density is rather flat toward the GC ($\mathcal{O}(2)$kpc core size), the wino signal is large enough to be detected by current experiments, and the thermal wino will be excluded under all plausible DM density models if no signal is seen in CTA.

This is an example of an indirect search probing regions of high-mass DM parameter space that cannot be explored by colliders (although the same models can be probed by other indirect searches, e.g. the antiproton searches discussed above). This scenario is also quite interesting from a theoretical perspective; in addition to the Sommerfeld enhancement, there are large double-log Sudakov corrections to the annihilation rate, which are most readily captured using methods of soft collinear effective theory (e.g. [107–114]). The closely-related higgsino model is not yet ruled out by any searches, although CTA may have the sensitivity to see a signal, depending on the DM density profile and high-energy backgrounds [106].

### 4.2.4 Annihilation of very heavy DM

For DM well above the TeV scale, constraints can be set using either gamma-ray telescopes (such as H.E.S.S, VERITAS, MAGIC, HAWC, and *Fermi*) or neutrino telescopes such as Ice-

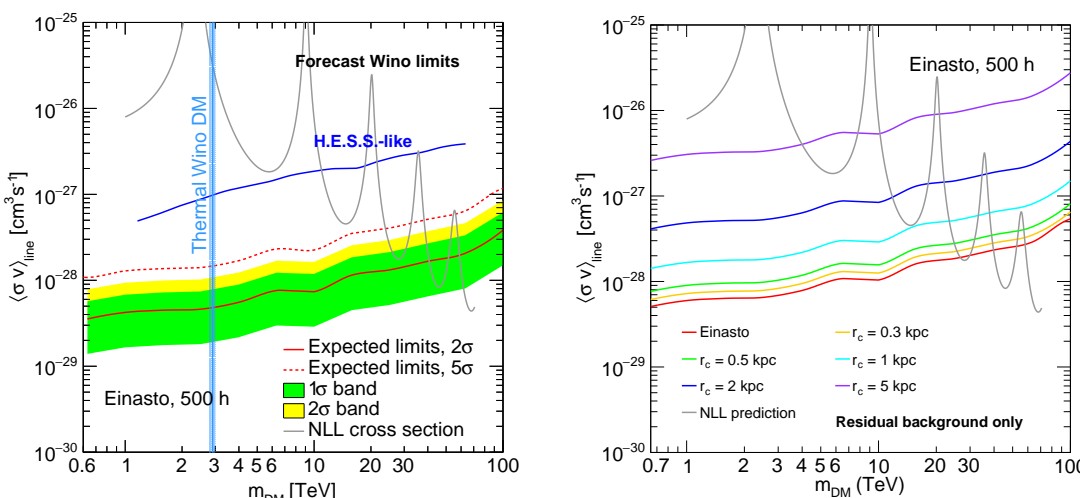

Figure 8: *Left panel:* The cross section for wino annihilation to produce a photon line, including the Sommerfeld enhancement and resummed logarithmic corrections (gray line), compared to the current sensitivity of H.E.S.S. (blue line) and the projected sensitivity of a 500h observation with CTA (red line), assuming an Einasto profile for the DM density. Green and yellow bands indicate the 1 and $2\sigma$ uncertainties on the expected $2\sigma$ bound from CTA. The vertical blue band indicates the DM mass range yielding the correct relic abundance. *Right panel:* the variation of the expected constraint from CTA if the DM density profile has a flat-density core at small Galactocentric radii. Reproduced from Ref. [106]; see that work for details.

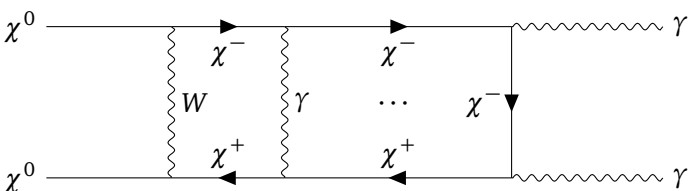

Figure 9: Example of annihilation of a supersymmetric wino $\chi^0$ to photons, via a long-range potential mediated by exchange of weak gauge bosons.

Cube and ANTARES. Fig. 10 shows some existing limits; the left panel is an older analysis, but the modeling of the signal includes contributions from DM substructure, and modeling of energy losses for gamma rays traveling intergalactic distances. Sufficiently high-energy photons can pair produce via interactions with the interstellar radiation field, producing an electron-photon cascade that results in a spectrum of gamma rays at lower energies; thus often observations from *Fermi*, which can observe photons in the 1-100 GeV range, are actually more constraining than from experiments that only observe higher-energy gamma rays. Note that the requirement of unitarity strongly constrains the annihilation rate at sufficiently high mass scales. The right panel shows recent constraints from observations of the Galactic Center by several gamma-ray and neutrino experiments, for DM up to 100 TeV [116]. In the future, the planned CTA and proposed SWGO telescopes have the potential to probe the thermal relic cross section for DM masses up to 10-100 TeV [117].

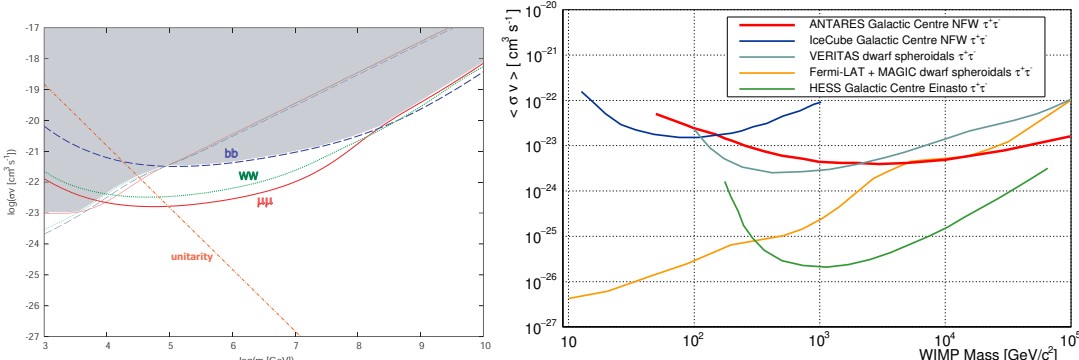

Figure 10: *Left panel*: neutrino (thick curves) and cascade gamma-ray (thin curves) constraints on the annihilation cross section of very heavy DM. The shaded region is determined by taking the more stringent of the neutrino and gamma-ray bounds, for the least constrained of the three channels. Reproduced from Ref. [115]. *Right panel*: updated limits for masses below 100 TeV for the sample channel of annihilation to $\tau$ leptons, using observations of the Galactic Center. Reproduced from Ref. [116].

### 4.2.5 Heavy dark matter decays

As for annihilation, heavy decaying DM can be constrained by observations from gamma-ray and neutrino telescopes. Fig. 11 summarizes the results of several analyses for the example $\bar{b}b$ channel; we see that generically lifetimes shorter than $\sim 10^{27-28}$ s can be ruled out, across a very large mass range. There are also limits on even heavier DM based on the non-observation of ultra-high-energy photons; Ref. [118] has a brief discussion of some current limits and future prospects.

### 4.2.6 Light dark matter annihilation and decay

Annihilation and decay of light DM, below the GeV scale, cannot be easily constrained by *Fermi*, for which effective area and angular resolution degrade rapidly at energies below a GeV. DM below the $\sim 100$ MeV scale also cannot decay into hadronic channels, which usually suppresses photon signals (due to lack of $\pi^0$ production), unless the DM decays directly to photons (which typically has a small branching ratio since the DM is uncharged). Similarly, sub-GeV DM annihilating or decaying into leptons is difficult to constrain with AMS-02, since AMS-02 is deep inside the Solar System and low-energy electrons are deflected by the solar wind.

The cosmological constraints discussed earlier, from the CMB and Lyman-$\alpha$ observations, remain important at low masses; for *s*-wave annihilation, the CMB bounds usually set the strongest constraints. Over most of parameter space, decays and *p*-wave annihilation of light DM are better constrained by observations of our present-day Galaxy, using data from lower-energy gamma-ray/X-ray telescopes, and electron spectrum measurements from the Voyager spacecraft. Voyager is a very old experiment, having been launched in the 1970s – but the designers did include a cosmic-ray spectrometer, and since 2012 the spacecraft has been outside the heliopause, allowing it to provide unique measurements of the cosmic-ray spectrum in interstellar space. This is particularly important for low-energy cosmic rays that would normally be deflected by the solar wind. Fig. 12 shows a selection of constraints on low-mass DM annihilating or decaying directly into photons; Fig. 13 shows similar bounds on low-mass DM annihilating or decaying to produce $e^+e^-$ pairs.

Combined with our results for higher masses, we observe that DM that decays visibly can

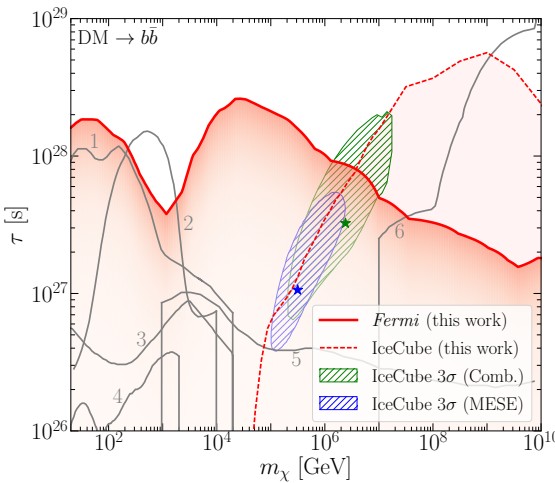

Figure 11: Lower bounds on the decay lifetime for DM decaying to $b$ quarks. The red line is determined by Ref. [56] using *Fermi* data; gray lines with numbers denote existing bounds using data from *Fermi* (2,3,5), AMS-02 (1,4), and PAO/KASCADE/CASAMIA (6). The hashed green (blue) region suggests parameter space where DM decay may provide a $\sim 3\sigma$ improvement to the description of the combined maximum likelihood IceCube neutrino flux. The red dotted line provides a limit if a combination of DM decay and astrophysical sources are responsible for the observed high-energy neutrinos. Reproduced from Ref. [56].

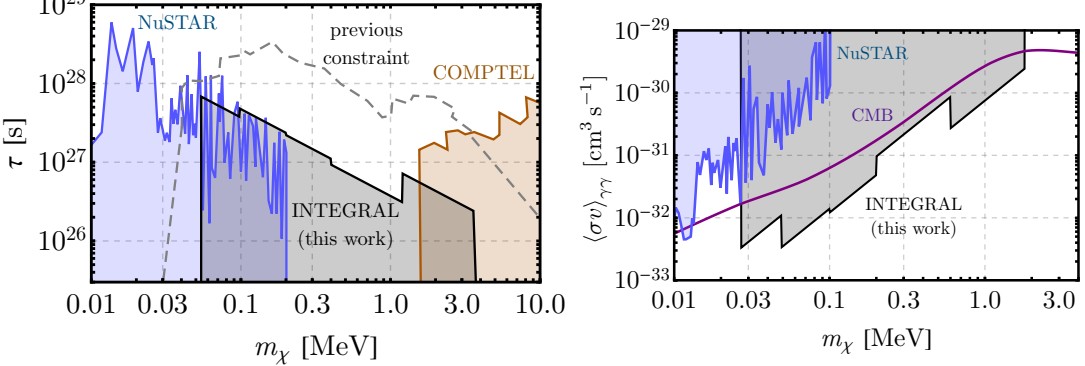

Figure 12: Selected bounds on the DM decay lifetime (*left panel*) and annihilation cross section (*right panel*) for annihilation/decay to $\gamma\gamma$, using data from the NuSTAR X-ray telescope, the CMB, and the INTEGRAL and COMPTEL gamma-ray telescopes. Reproduced from Ref. [119]; see that work for further details. A compilation of bounds at lower/higher masses can be found in Ref. [120].

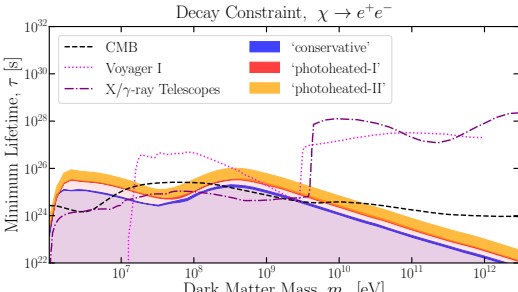
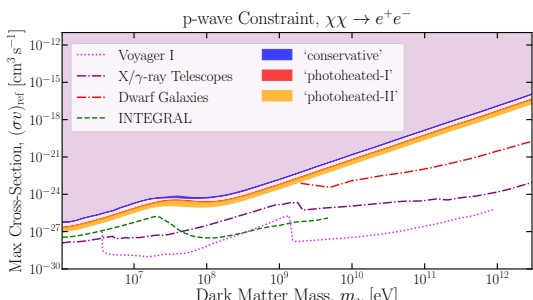

Figure 13: Selected bounds on the DM decay lifetime (*left panel*) and annihilation cross section (*right panel*) for annihilation/decay to $e^+e^-$. The blue and red lines and orange bands represent constraints derived from the Lyman-$\alpha$ forest under various assumptions for the astrophysical background; the black dashed line indicates the CMB bound (driven primarily by excess early ionization); the pink dotted line indicates limits on low-energy cosmic rays from Voyager I; green-dashed and purple dot-dashed lines indicate limits from X-ray and gamma-ray telescopes. Reproduced from Ref. [71]; see that work for further details.

be constrained over a very wide mass range (keV-EeV) to have a lifetime longer than $\sim 10^{27-28}$ s. The bound is somewhat weaker for sub-10-GeV DM decaying primarily leptonically, which can have a lifetime as short as $\sim 10^{25}$ s. However, in all cases, only a very tiny fraction of DM can decay over the lifetime of the universe. These limits also have implications for scenarios where DM is composed of primordial black holes and decays through Hawking radiation.

Proposed future gamma-ray telescopes have the potential to significantly improve our sensitivity to gamma rays in the MeV band. A discussion and initial sensitivity estimates for several proposed missions are given in Ref. [121]. Recent whitepapers on proposed missions are available in many cases, e.g. AMEGO [122], e-ASTROGAM [123], GECCO [124], and GRAMS [125].

## 4.3   Exercise 3: detectability of gamma-ray lines from the Galactic Center

- For the Burkert, Einasto ($\alpha = 0.17$) and NFW profiles, compute the *J*-factor (for annihilation) from the region between 1 and 3 degrees from the Galactic Center. For the NFW case, also consider the *generalized* NFW profile, $\rho(r) \propto r^{-\gamma}/(1 + r/r_s)^{3-\gamma}$, for inner slope $\gamma = 0.5$ and 1.5. Take the scale radius to be 20 kpc for the NFW and Einasto profiles, and 10kpc for the Burkert profile. You may assume the Earth is 8.5 kpc from the Galactic Center, and take the local dark matter density at the position of the Earth to be 0.4 GeV/cm$^3$. Give all results in GeV$^2$/cm$^5$.

- Imagine you have 130 GeV DM annihilating with a cross section of $\langle \sigma v \rangle \approx 10^{-27}$ cm$^3$/s to a two-body final state of $\gamma Z$, and with equal cross section to a two-body final state of $\gamma\gamma$. What are the energies of the two resulting gamma-ray lines?

- What total photon flux (integrating over energy) associated with these lines do you predict from the region in (a), as measured at the Earth, given the parameters in (b)? Assume the NFW profile with $\gamma = 1$ and give your answer in photons per square centimeter per second. For an instrument with an effective area of one square meter, how many line photons do you expect to see from this region over the course of four years?

  (There was a claimed detection of just such a line in 2012, although its significance faded away with time.)

If you like, as a bonus question, you can also compute the photon and charged-particle spectra associated with the decay of the *Z* and estimate the corresponding signals at Earth. How does the typical energy of the photons from *Z* decay compare to the line photon energies?

## 5  Lecture 5: some current anomalies and excesses

In addition to these limits, there are a few channels in which possible DM signals have been claimed. Let us now summarize the status of a few of these.

### 5.1  Cosmic ray positrons

By the arguments given in section 3.1, the ratio of secondary to primary cosmic rays is generally expected to decrease with increasing energy. However, a rise in the positron fraction – the ratio of positron flux to total electron+positron flux – at energies above $\sim 10$ GeV was observed by the PAMELA experiment in 2008 [126], and has since been confirmed by *Fermi* [127] and AMS-02 [128]. The AMS-02 measurement has much smaller uncertainties than the original PAMELA detection, and extends to higher energies; the positron fraction appears to continue to rise up to energies of $\sim 300$ GeV. At higher energies, there is some evidence of a turnover, but the statistical uncertainties are large.

One possible explanation for this excess is DM annihilation or decay, producing additional primary positrons. Other possibilities include positrons sourced by pulsars (e.g. Ref. [129]), secondary positrons re-accelerated in some way (e.g. inside supernova remnants) [130], or some substantial modification to our understanding of cosmic-ray production or propagation [131]. Under the DM-origin hypothesis, the DM must be quite heavy, with mass at least several hundred GeV; the annihilation or decay must occur primarily into leptonic channels to avoid constraints from antiproton and gamma-ray searches; and if annihilation is responsible, the cross section must be well above the thermal relic value (e.g. Ref. [64]). While all of these features can be found in DM models [132], the required parameters are in tension or apparently excluded by the constraints discussed above (e.g. the annihilation explanation is generically in tension with CMB bounds [5], and the decay explanation with limits from gamma-ray observations [56,133,134]). It may still be possible to explain the excess with DM models with additional features, e.g. annihilation to a long-lived mediator that subsequently decays, especially combined with large localized DM overdensities; however, the DM densities suggested by these explanations are often much larger than expected based on simulations (see Ref. [135] for an example model).

Anisotropy in cosmic-ray arrival directions could potentially probe the distribution of the positron sources. However, Galactic magnetic fields scramble the arrival direction, and consequently the expected anisotropy is small, below the 1% level, even if the source is a single nearby pulsar (e.g. Ref. [136]). Current measurements by *Fermi* [137] and AMS-02 [128] find no evidence for anisotropy, but are not sensitive to such small anisotropies in any case. However, more sensitive anisotropy measurements could be obtained using observations of cosmic rays by atmospheric Cherenkov telescopes [136]; while designed to observe high-energy gamma-rays, these telescopes are sensitive to cosmic-ray collisions with the atmosphere (in fact this is their main background).

The pulsar hypothesis has gained traction in recent years, due to the discovery in HAWC data of TeV gamma-ray halos surrounding nearby pulsars [138]. These gamma-ray halos are thought to originate from inverse Compton scattering by high-energy electrons and positrons streaming off the pulsar. Their existence thus indicates that these pulsars are accelerating a substantial number of electrons and positrons to multi-TeV energies, which is one of the major

requirements to explain the excess. However, the fact that these halos are observable suggests that diffusion must be modified in the neighborhood of pulsars [138–140]. The standard estimates for diffusion length would imply that $e^+e^-$ pairs with energy sufficient to generate the gamma-rays observed by HAWC should diffuse $\mathcal{O}(100)$ parsecs, and so for the nearest pulsars which are only $200-300$ parsecs away, their $e^+e^-$ halos should extend tens of degrees across the sky. In actual fact their extension is only a few degrees, suggesting impeded diffusion around the pulsars – but we know this impeded diffusion cannot be ubiquitous, because otherwise we would not observe the spectra of cosmic-ray electrons that we do at Earth (given we expect the closest sources of high-energy electrons to also be a few hundred parsecs away) [141]. Depending on the model for impeded diffusion, nearby pulsars may or may not be able to explain the positron excess, but plausible models allow for it (e.g. [139, 140, 142]).

## 5.2 Cosmic ray antiprotons

Several independent groups of theorists have claimed a modestly significant excess in AMS-02 data for antiprotons with energies around 10-20 GeV [143,144]. If interpreted as a DM signal, this signal would suggest $\mathcal{O}(40-130)$ GeV DM annihilating with a roughly thermal relic cross section. The initial reported significance of the excess was around $4\sigma$.

However, there has been considerable debate in the literature on the true significance of this apparent excess. There are significant systematic uncertainties in both the background and signal models, arising from uncertainties in the propagation model, the model for astrophysical sources, and the cross-sections for antiproton production and interaction in the detector. Furthermore, the treatment of possibly correlated errors in different energy bins can markedly affect the apparent significance of the result.

For example, Refs. [145, 146] both sought to estimate plausible covariance matrices for the data, describing correlations between the uncertainties for different data points, based on theoretical modeling and consistency with other AMS-02 results. Ref. [145] found the data were fully consistent with an astrophysical origin, while Ref. [146] found the significance of the excess instead increased to over $5\sigma$. Ref. [147] agreed with Ref. [146] in finding a high significance after taking a number of systematic uncertainties into account. A later study focused on correlated systematic uncertainties in the absorption cross-section of cosmic rays in the detector material [148], and argued that using a detailed theoretical model for these correlations and a prescription for other systematic uncertainties reduces the significance of the excess to below $1\sigma$.

## 5.3 Cosmic ray anti-helium

The AMS-02 experiment has reported the tentative possible detection of six apparent anti-He-3 events and two apparent anti-He-4 events [149]. This was very unexpected and would be quite surprising if confirmed, as both the expected astrophysical background and the expected signal from new physics (such as DM annihilation to quarks, with cross sections that are not currently excluded) are well below AMS-02's sensitivity.

Two new physics proposals for explaining these events are (1) clouds of antimatter or "anti-stars" in the Milky Way Galaxy [150], (2) DM annihilation, but with enhanced anti-helium production via a process that is not properly captured by the standard event generators `Pythia` and `Herwig` [151]. Specifically, the authors of Ref. [151] argue that production of $\bar{\Lambda}_b$-baryons has been underestimated by these generators. Such baryons have quark content $\bar{u}\bar{d}\bar{b}$, and mass 5.6 GeV; consequently, they are kinematically permitted to decay producing an anti-helium nucleus.

The standard method for estimating the antinucleon yield in a reaction is based on a "coalescence model", where antiparticles produced with relative momentum smaller than some

threshold value are assumed to coalesce into heavier antinuclei. The value of the threshold momentum is tuned to match data from colliders. The authors of Ref. [151] argue that the standard methods systematically discount contributions from production of long-lived particles that travel some distance and subsequently decay to antinucleons.

It is not yet clear if this mechanism will allow a rate high enough to fully explain the AMS-02 events; Ref. [151] attempted to recalculate the rate taking these displaced decays into account, but found that `Pythia` and `Herwig` gave quite different results, and ascribed this to differences in the underlying hadronization models. Consequently we may need theoretical improvements to accurately assess whether this is a viable explanation. (There has been some further discussion of the event generator predictions on the arXiv, see Refs. [152, 153].)

## 5.4 The 3.5 keV line

An apparent spectral line at an energy of 3.5 keV was discovered in XMM-Newton observations of galaxy clusters in 2014 [154, 155]. The significance of the signal was (at the time) roughly $4\sigma$. A review of this potential signal and its possible interpretations has been given by Ref. [156]; here let us summarize some key points.

A large number of follow-up observational studies have been performed; the signal has now also been detected tentatively in the Galactic Center [157] and the cosmic X-ray background [158]. Observations of the Draco dwarf galaxy have yielded mixed results, with claims of both a faint signal and a strong exclusion [159, 160]. A recent survey of galaxy clusters using XMM-Newton data found excesses in some clusters but not in others, disfavoring interpretations where the signal traces the total DM content [161]. Studies of M31 with Chandra [162], stacked galaxies with Chandra and XMM-Newton [163], dwarf galaxies with XMM-Newton [164], blank-sky observations with XMM-Newton [165, 166], and observations of the Milky Way DM halo with archival Chandra data [167] and with HaloSat [168], have not found a signal and have instead set stringent limits. However, some of these limits are hotly debated. (As one example, you can see how the analysis of Ref. [165] works yourself at https://github.com/bsafdi/BlankSkyfor3p5, and read the back-and-forth on the arXiv in Refs. [169–171].)

The simplest DM-related explanation is a decaying sterile neutrino with a mass around 7 keV. The sterile neutrino is a long-standing DM candidate, and if its mass is above a few keV, it can be sufficiently cold to evade constraints on warm DM (e.g. [156] and references therein). However, the simple DM decay model is very predictive: the signal from any body should be proportional to the total amount of DM in that body. This hypothesis appears to be in some tension with the null results described above. There is also a (disputed) claim in the literature that the spatial morphologies of the signals observed from the Galactic Center and Perseus cluster are incompatible with decaying DM; see Refs. [156, 172] for competing viewpoints.

There are several alternative DM-related possibilities that are less predictive, and hence less constrained. One example is "exciting DM" [173, 174]; in this scenario the DM itself may be much heavier than the keV scale, but it has a metastable excited state 3.5 keV above the ground state. This state is excited by DM-DM collisions, and subsequently decays producing a 3.5 keV photon. The rate of excitation would scale as the DM density squared, with some non-trivial velocity dependence; this combination of parameters is far less constrained than the total DM content of an object, and could e.g. explain why no signal is seen in dwarfs (the typical DM velocity is too low to excite the excited state) while appearing in galaxy clusters (where the typical DM velocity is much higher). Another, independent, possibility is that the DM might decay producing a 3.5 keV axion-like particle (ALP), which could convert to a 3.5 keV photon in an external magnetic field [175]; the signal would then depend on the ambient magnetic field, leading to widely varying signals from different systems [176].

There is an ongoing debate over possible contamination from potassium and chlorine plasma lines; a spectral line at a few keV is much easier to mimic with atomic processes than

a gamma-ray line (see e.g. Refs. [177–180] for discussion). There are several known X-ray lines close to 3.5 keV and their strength can depend sensitively on the plasma temperature. Charge-exchange reactions have been experimentally verified to produce 3.5 keV emission, and may give rise to some or all of the observed feature [181, 182] (see also the discussion in Ref. [158], which attempts to exclude this explanation).

Energy resolution may be the key to distinguishing between DM and astrophysical origins for the line. An instrument with sufficiently good energy resolution could potentially resolve the putative 3.5 keV line at an energy distinct from any of the known atomic lines; with eV-scale energy resolution, it would be possible to measure the Doppler broadening due to the velocity of DM in the Galactic halo, if the signal originates from DM decay. The Hitomi telescope had the required energy resolution, but failed a few days after launch; a short observation of the Perseus cluster [183] did not find evidence for a $\sim$ 3.5 keV line, in tension with earlier measurements claiming a bright line in Perseus, but the limits on the signal strength do not probe the DM decay explanation (fully explaining the claimed bright signal in Perseus is challenging in the context of DM decay).

There are a number of prospects for probing the 3.5 keV line with upcoming experiments:

- The eROSITA X-ray telescope was launched in 2019 and is currently performing an all-sky survey. Its sensitivity after four years should be sufficient to detect or exclude the line [184], although its energy resolution ($\sim$ 120 eV) will not be good enough to resolve the shape of the line.

- eXTP is an X-ray timing and polarimetry mission with a target launch date of 2025. While it is not designed for DM searches, its large field of view and effective area should give it excellent sensitivity to sterile neutrinos [185], although again it will not have the energy resolution to resolve the lineshape.

- The Micro-X sounding rocket program [186, 187] offers the possibility of eV-scale energy resolution in the relatively near term. By placing high-resolution X-ray spectrometers on suborbital sounding rockets, this approach would achieve excellent energy resolution – as low as 3 eV – for modest cost. The exposure would be short – 5 minutes – and there would be essentially no pointing information, but the instrument's field of view would be large, with roughly a 20 degree radius. The strategy would be to search for a DM decay signal from the local Galactic halo, rather than from localized targets such as galaxy clusters and the Galactic Center; Micro-X should have the sensitivity to observe the line even with such short flights. Micro-X flew an initial flight (not for a DM search) in 2018.

- XRISM is also expected to have few-eV energy resolution and is scheduled for launch in 2022; velocity spectroscopy in the Milky Way will become possible with this instrument, but it will require very long exposure times, 4 Ms and higher [188].

- There are other instruments with even better energy resolution planned for launch in the 2030s, Athena and Lynx, which will also have much better sensitivity.

## 5.5 21cm absorption

In March 2018, the Experiment to Detect the Global Epoch-of-reionization Signature (EDGES) reported an apparent detection of a deep primordial absorption trough in redshifted 21cm radiation, corresponding to redshifts 15-20 [67]. The relative amount of 21cm emission vs absorption is determined by the comparison of the radiation temperature $T_R$ (or more precisely, the photon intensity at the relevant wavelength) to the "spin temperature" of the hydrogen gas. The spin temperature $T_s$ describes the relative abundance of the ground and excited

states of the 21cm hyperfine transition; the ratio of abundances is the equilibrium ratio at the spin temperature. If the radiation temperature exceeds the spin temperature, there will be net absorption; in the opposite case, net emission. More generally, the 21cm brightness temperature (governing the intensity of the line) is proportional to $1 - T_R/T_s$, as well as to the fraction of hydrogen that is ionized [8].

EDGES's detection of a deep absorption trough thus suggested $T_s < T_R$ at $z \sim 15-20$, and set an upper limit on $T_s/T_R(z = 17.2) < 0.105$ at 99% confidence. This is surprising because:

- The expectation from standard cosmology is that $T_s$ should lie between the radiation temperature and the gas kinetic temperature $T_{\mathrm{gas}}$, with $T_s$ becoming tightly coupled to $T_{\mathrm{gas}}$ through the Wouythusen-Field effect once UV photons from stars are abundant (see e.g. [189] for a discussion); in any case we should have $T_s \geq \min(T_R, T_{\mathrm{gas}})$.

- The gas temperature has a non-zero primordial value because the free electrons remained efficiently coupled to the photon bath down to $z \sim 150-200$, and this kept the gas at the same temperature as the CMB. The physics here is electron-photon scattering and is very well-understood. After these temperatures decouple, the gas cools as a non-relativistic species, with its momentum redshifting as $1/a$ and its kinetic energy (and temperature) redshifting as $1/a^2$; in contrast, the CMB temperature continues to cool proportionally to $1/a$. Thus the gas temperature has a minimum value set by its primordial heating, given roughly by $T_{\mathrm{gas}} = T_{\mathrm{CMB}}(1+z)/(1+z_{\mathrm{dec}})$, where $z_{\mathrm{dec}} \sim 150-200$ is the redshift at which the temperatures of the gas and the CMB first decoupled. The gas may be heated to higher temperatures by photons from stars, but it is challenging to see how it could be appreciably cooled below this minimum value via SM physics.

- Assuming standard cosmology, we can predict this minimum gas temperature to be $T_{\mathrm{gas}} \approx 7$ K at $z = 17.2$. Assuming $T_R$ is the CMB temperature, the EDGES result implies that at $z = 17.2$:
$$T_{\mathrm{gas}} < T_s < 0.105 \, T_{\mathrm{CMB}} = 5\mathrm{K}. \tag{18}$$

The claimed absorption depth is thus inconsistent with the standard picture. Furthermore, in a realistic scenario, perfect coupling between the spin and gas temperatures should require some significant flux of photons from stars, which should in turn lead to photoheating of the gas – thus even a saturation of this limit would be surprising, and a violation suggests a breakdown of our assumptions.

It is of course plausible that this result is due to an issue with the experimental setup, or a feature in the foregrounds; this is a very difficult measurement and the first claim of such a primordial signal. Refs. [190, 191] discuss some possible sources of systematic error and their ability to explain the apparent signal. If this observation is truly a measurement of the primordial 21cm radiation, that suggests that either $T_R$ is larger than expected (suggesting a new source of early radiation), $T_{\mathrm{gas}}$ is smaller than expected (suggesting a new cooling mechanism; scattering between the baryons and a small sub-component of the DM is a possibility), or some other change to standard cosmology is required. The confirmation of such a deep absorption trough, and understanding its origin, could also allow us to set particularly stringent constraints on DM annihilation and decay during this epoch, as they would act to heat the gas and hence reduce the amount of absorption [192].

One possibility that gained particular interest, in the second category, is the idea that perhaps some small fraction of the DM carries a tiny electric "millicharge". Rutherford scattering between such millicharged DM and visible charged particles would have a cross section proportional to $v_{\mathrm{rel}}^{-4}$, allowing for very large enhancements to the scattering rate at low velocities. This large scattering rate would help transfer energy from the (hotter) visible matter to the

(colder) DM – the DM would effectively act as a heat sink for the visible matter. (This scenario has at least two major variations, depending on whether the millicharged component interacts with the rest of the DM; see e.g. Ref. [193].) However, the available parameter space is very limited, at least when the fraction of the DM that is not millicharged is assumed to be inert and to have no interactions with the millicharged component (other than gravitational) [194, 195]. In this case, the DM must have a mass between 0.5-35 MeV, and the fraction of DM that is millicharged must lie between $(m_{\text{DM}}/\text{MeV})0.0115\%$ and $0.4\%$ of the total. If the millicharged component interacts with the neutral component, however, much smaller millicharged fractions can efficiently cool the baryons. Direct detection experiments searching for light DM could potentially also probe the millicharged component, although one concern is the possibility that millicharged DM could be evacuated from the neighborhood of the Earth by outflows from the Galactic disk and interactions with the Galactic magnetic field [196–198].

### 5.6 Exercise 4: dark matter velocity spectroscopy

Suppose a 7.0 keV sterile neutrino decays into a neutrino and a photon in the DM halo of the Milky Way. You can treat the neutrino as massless for purposes of this problem. The photon travels to Earth and is detected by an X-ray telescope there.

- Suppose the sterile neutrino has velocity $\vec{v}$ in the frame of the Earth prior to its decay. For simplicity, assume that this velocity vector points (a) radially away from the Earth or (b) radially toward the Earth. What is the energy of the photon as observed at Earth, as a function of $v$, in both cases? You may assume $v$ is very small compared to the speed of light.

- Now consider the more realistic case where the sterile neutrinos are approximately at rest relative to the Galaxy, and the Earth is traveling on a circular orbit around the Galactic Center with speed $v$. From the perspective of the Earth, suppose a given sterile neutrino is at position $(l, b)$ in Galactic coordinates at the time it decays. How does the line-of-sight velocity of the sterile neutrino relative to Earth depend on $l$ and $b$? Again, you may assume $v$ is very small compared to the speed of light. (Note this question is about trigonometry, not physics.)

- Suppose the circular velocity of the Earth is 220 km/s, and we perform two observations, in one case pointing my telescope at $(l, b) = (30°, 30°)$, and in the other case at $(l, b) = (-30°, 30°)$. What is the difference in energy you expect to observe between photons from the two locations, if they originate from sterile neutrino decay? What relative energy resolution would my instrument need in order to distinguish the two?

## 6 Lecture 6 (bonus): a case study of the Galactic Center GeV excess

### 6.1 Modeling continuum gamma-ray backgrounds

For continuum gamma-ray signals, as opposed to lines, the Galactic Center is a challenging region due to large astrophysical backgrounds; to proceed, we need a way to estimate or parameterize these backgrounds. At weak-scale energies, the dominant backgrounds come from:

- Cosmic ray protons striking the gas, producing neutral pions which decay to gamma rays.

- Cosmic ray electrons upscattering starlight photons to gamma-ray energies.

- Compact sources producing gamma-rays – pulsars, supernova remnants, etc.

While the underlying physical processes generating these backgrounds are largely well-understood, the three-dimensional distributions of gas, starlight and cosmic rays are not well-measured, making precise prediction difficult. However, we can at least say that these backgrounds should roughly trace the distributions of gas and stars (stars can be gamma-ray point sources themselves, or generate starlight that is upscattered by cosmic-ray electrons; supernovae are thought to generate cosmic rays), which are much more dense in the disk of the Milky Way.

A model for the astrophysical diffuse (non-point-source) backgrounds can be constructed from maps of the gas distribution and models for the cosmic-ray and radiation distributions; for example, the latter can be taken from the public GALPROP code. There are existing public models made available by the *Fermi* Collaboration [201, 202];[3] however, caution should be used when employing these models for analysis of diffuse signals, as they are designed for point source searches. The later official *Fermi* models either include ad hoc spatial templates to absorb large-scale discrepancies between the initial model and the data, or re-add smoothed data-minus-model residuals (which would include any large-scale diffuse signals) to the model at the final step of processing.

Typically one models the sky as a linear combination of spatial "templates", each corresponding to one or more emission mechanisms; the normalization of these templates may be fitted separately in each energy bin, or a spectrum for the template may be imposed externally and only the overall normalization fitted. This approach is not restricted to gamma-rays; similar template methods have been used in the microwave sky to remove foregrounds in studies of the CMB (e.g. Ref. [203]), and to probe possible DM signals (e.g. Refs. [204, 205]).

Fig. 14 displays an example of such a fit using an early *Fermi* Collaboration diffuse model (note the disk-like distribution of emission, brightest along the plane of the Milky Way) and a simple template for the large-scale gamma-ray structures known as the *Fermi* Bubbles [206], divided into slices by latitude. The normalization of each template is allowed to float in each energy bin, allowing the extraction of a data-driven spectrum for each model component, as shown in the figure. This analysis was used in Ref. [200] to study the spectrum of the *Fermi* Bubbles as a function of latitude; the pronounced GeV-scale bump in the lowest-latitude slice is associated with the Galactic Center GeV excess, which we will discuss next.

## 6.2 Properties of the GeV excess

Such template-based studies indicate the presence of a new GeV-scale gamma-ray emission component in the Galactic Center (initially found by Refs. [207, 208]), and the inner Galaxy within $\sim 10°$ of the Galactic Center (initially found by Ref. [200]). The spectrum of this component is peaked at 1-3 GeV, and if interpreted as a possible DM signal, the size and spectrum of the signal are consistent with relatively light ($\lesssim 100$ GeV) thermal relic DM annihilating to quarks. Spatially, the signal resembles the square of a slightly steepened NFW profile (inner slope of $r^{-1.25}$ rather than $r^{-1}$), with no flat-density core; that is, it would be consistent with DM annihilation from a slightly steepened NFW profile, or with an astrophysical source population with a power-law slope around $r^{-2.5}$. The existence and general properties of this excess have been confirmed in studies by the *Fermi* Collaboration [209], as well as work by a large number of independent groups. This component is frequently referred to as either the "GeV excess" or the "Galactic Center excess" (GCE), despite its extension throughout the inner Milky Way; I will adopt the "GCE" label here.

---

[3]Current and previous models are available from the *Fermi* website, https://fermi.gsfc.nasa.gov/ssc/data/access/lat/BackgroundModels.html

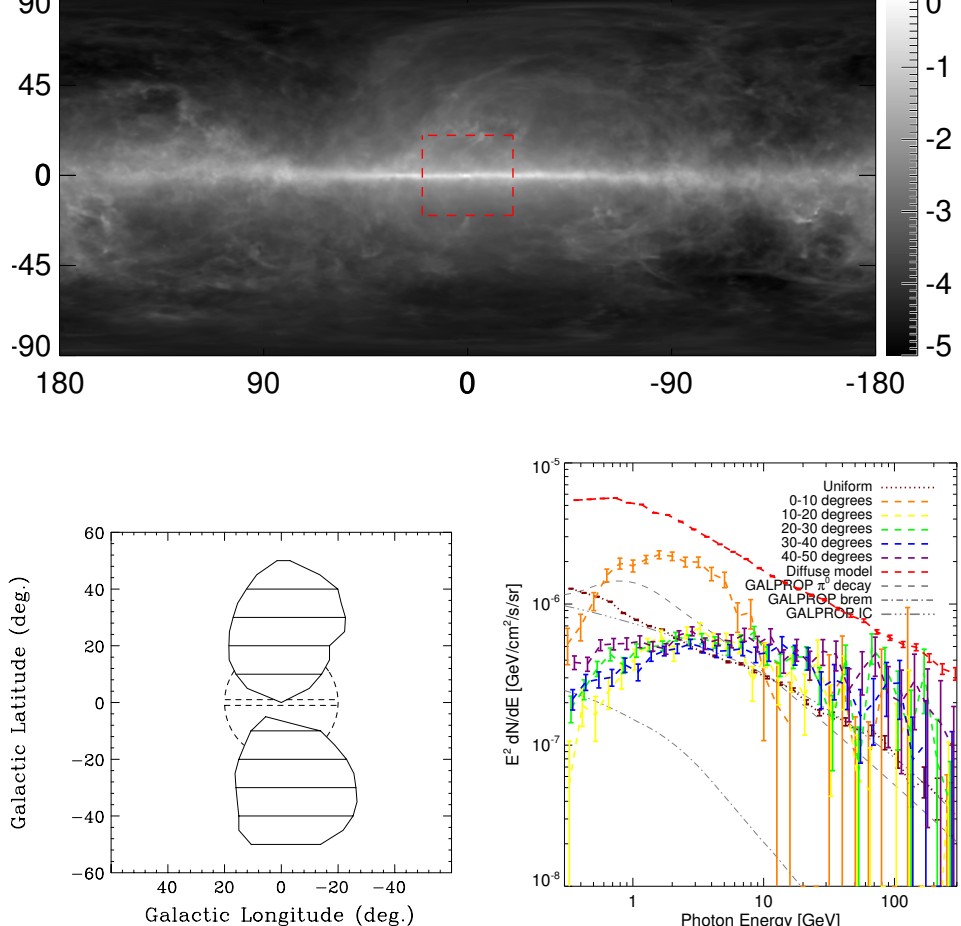

Figure 14: *Top:* `p6v11` *Fermi* Collaboration model for the diffuse gamma-ray emission, evaluated at an energy of 1 GeV; the *x* and *y* axes denote Galactic latitude and longitude respectively, and the red dashed lines indicate a $40 \times 40°$ region around the Galactic Center (reproduced from Ref. [199]). *Bottom left:* spatial template for the *Fermi* Bubbles. Horizontal lines indicate ten-degree bands in Galactic latitude. Dashed lines indicate the region within 20° of the Galactic center, but more than 1° from the Galactic plane. *Bottom right:* Spectra extracted from a template fit, modeling the sky as a linear combination of the diffuse model shown in the top panel (red dashed line), the isotropic gamma-ray background (brown dotted line), and the latitudinally sliced templates for the *Fermi* Bubbles shown in the bottom *left panel* (orange, yellow, green, blue and purple dashed lines, for latitudes of $0 - 10°$, $10 - 20°$, $20 - 30°$, $30 - 40°$, $40 - 50°$ respectively). Gray lines indicate the expected spectra from cosmic rays interacting with the gas and starlight. Reproduced from Ref. [200]; see that work for details.

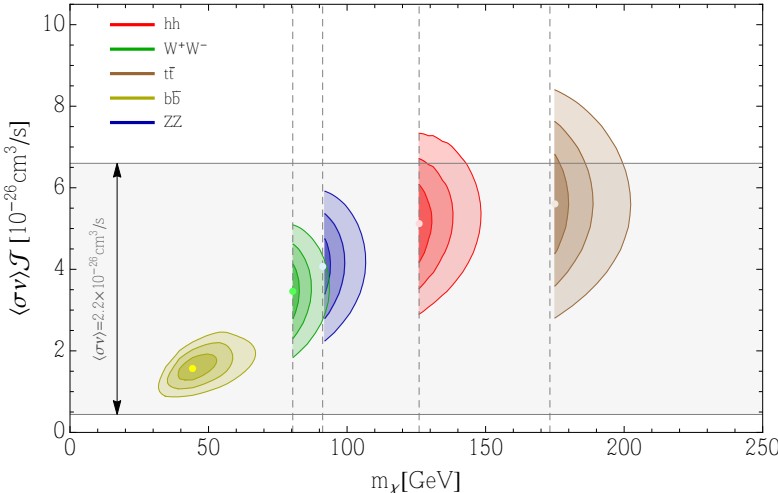

Figure 15: Preferred regions in cross section and mass for DM annihilation to various SM final states, if DM annihilation explains the full Galactic Center excess. Reproduced from Ref. [215].

The morphology of the GCE is fairly spatially symmetric about the Galactic Center, rather than strongly elongated like the Galactic disk [199, 210]; the central peak also appears to be centered on the Galactic Center [199]. This symmetry is suggestive of an origin in the halo of the Milky Way, rather than the disk. However, it is not clear at present if the GCE is truly spherically symmetric, or instead more closely resembles the stellar bulge of the Milky Way (which is not nearly as elongated as the Galactic disk, but still distinctly non-spherical).

Early studies [199, 210] using Galactic diffuse emission models based on GALPROP found that a spherical morphology was preferred, in comparison to tests where the excess template was stretched along some arbitrary axis. A recent study [211], using updated models for the Galactic diffuse emission (albeit still based on GALPROP), found very similar results. That work also explicitly tested a model for the GCE spatially matching the Galactic bulge emission, and found it provided a significantly worse overall fit compared to the spherically symmetric (steepened) NFW model.

However, studies using alternative approaches to model the Galactic diffuse emission have found that using their background models, a template resembling the stellar bulge provides a significantly better description of the excess than a spherical model [212–214]. If robustly confirmed, this would be an important result, suggesting some kind of stellar origin for the GCE.

## 6.3 Implications of a DM origin

If the excess originates from DM, the greatest improvement in the fit occurs for DM masses around 10-100 GeV depending on the annihilation channel. For $b$ quarks, the overall best-fit channel, the best-fit mass is $\sim 40-50$ GeV. The required cross section is close to thermal, i.e. approximately weak-scale. Heavier DM annihilating to $\bar{h}h$ can also provide a good fit [215], but the preferred DM mass is right at the threshold for $\bar{h}h$ production. Annihilation to W bosons, Z bosons and top quarks provides a slightly worse fit; again the preferred mass is close to threshold, as shown in Fig. 15.

There are non-negligible model-building challenges for a DM explanation of this signal, although there are existence proofs of UV-complete models that satisfy all constraints. Direct detection is very sensitive in this mass range, so the lack of a detection must be explained.

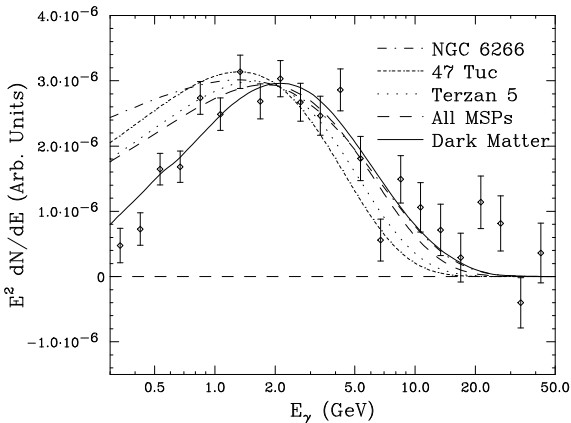

Figure 16: Observed spectrum of the GeV excess in the analysis of Ref. [199], compared to spectra of observed millisecond pulsars and several globular clusters (whose gamma-ray emission is thought to be dominated by millisecond pulsars. Reproduced from Ref. [199]; see that work for details.

Some possibilities include a resonant enhancement to the annihilation rate, a suppression of the direct detection rate due to spin-dependence or some other effect (although upcoming direct-detection experiments may have sensitivity anyway), or a scenario where the annihilation is to intermediate particles that subsequently decay into visible particles with a small coupling. In this third case, the small coupling between the dark sector and SM suppresses the direct detection rate, but not the annihilation rate.

There are also important limits from colliders, ruling out substantial classes of simplified models [216]. The sensitivity of collider searches is reduced in the presence of light mediators, which may be needed to raise the cross section to thermal relic values.

Two example classes of viable models are:

- Annihilation through a pseudoscalar to *b* quarks, e.g. the "coy DM" of Ref. [217]. A renormalizable model, where the pseudoscalar mixes with the CP-odd component of a two-Higgs-doublet model, was presented in Ref. [218]. An implementation in the $Z_3$ NMSSM was worked out in Ref. [219], where bino/higgsino DM annihilates through a light MSSM-like pseudoscalar. A general NMSSM study was performed in Ref. [220].

- $2 \to 4$ models, where the DM annihilates to an on-shell mediator, which subsequently decays to SM particles, e.g. Refs. [221–223]. Dark-photon and NMSSM implementations are discussed in Ref. [224]; an extension with dark-sector showering is presented in Ref. [225].

## 6.4 Non-DM possibilities

Pulsars, spinning neutron stars, are known to emit gamma rays with a very similar spectrum to the observed excess, as shown in Fig. 16. There have been proposals for generating an abundant pulsar population in the stellar bulge that does not violate limits from the non-observation of bright pulsars [226, 227], or a pulsar population with a spherical distribution peaked toward the Galactic Center [228].

Outflows of high-energy cosmic rays from the Galactic Center could also produce gamma rays, as discussed previously. However, it would be surprising if the excess originated wholly from protons interacting with the gas, as the signal does not appear to be gas-correlated. Electrons upscattering photons to gamma-ray energies could also contribute, although multiple

sources may be required to accommodate an electron spectrum that does not change markedly with position. For discussion of these points and more, see e.g. Refs. [229–232].

## 6.5 Clues from the GCE morphology

As mentioned above, the GCE has been claimed to possess both a spherical and a bulge-like morphology. The latter would be a strong hint toward some kind of origin in the physics of the Milky Way's stellar bulge, and in particular could be a clue that we are looking at a novel and previously-unexpected pulsar population. A spherical morphology would not exclude the pulsar explanation, but would be more easily explained by a DM origin (the DM halo is expected to be quite spherical close to the GC, e.g. [233]). Let us now go into a bit more depth on this debate.

There are broadly two types of background models that have been demonstrated to prefer a bulge-like morphology. The method of Refs. [213, 214] also uses GALPROP to estimate the inverse Compton scattering to the Galactic diffuse emission, but for the gas-correlated components, these studies use modified maps of the interstellar gas derived from hydrodynamical simulations. The resulting gas maps were found to provide a better fit relative to the GALPROP default approach at the time of writing of Ref. [213] (the author is not aware of a formal test of whether they are a better fit compared to more recent GALPROP-based models, as in Ref. [211]).

Perhaps more importantly, neither these models nor GALPROP-based models provide a formally *good* description of the data in a statistical sense (even once a model for the GCE is added). Consequently, there are certainly systematic uncertainties associated with the differences between even the best-fitting Galactic diffuse emission models and the truth. The question of how to assess those systematic uncertainties, and how they affect the inferred properties of the GCE, is the major obstacle to a full understanding of the signal.

The second type of background model is based on the SkyFACT method developed in Ref. [234]. SkyFACT begins with a standard template analysis, but then allows certain templates to vary on a pixel-to-pixel basis in order to provide a better description of the data, effectively adding a very large number of nuisance parameters. Regularization terms in the likelihood ensure spatial and spectral smoothness of the resulting templates, and typically templates with extra parameters to describe spatial variation are required to have a consistent energy spectrum (that is, the modulation of the template, at a given location and energy, fixes the modulation at the same location for all energies).

To justify this approach, consider the emission from cosmic-ray protons interacting with the gas, which is the largest source of Galactic diffuse gamma-ray emission at the relevant energies. The proton spectrum is expected to be fairly stable across large regions of the Galaxy, but the spatial structure in the gamma-rays relies on the 3D distribution of gas density, which is likely not well-captured in our current models and can lead to discrepancies between the models and the gamma-ray data. To fix these discrepancies, we can allow the gas column density to vary away from its presumed value on a pixel-to-pixel basis, while requiring that any emission attributed to this component has the correct spectrum (i.e. the spectrum of gas-correlated emission elsewhere in the Galaxy).

This approach brings the model much closer to being a statistically well-fitting description of the *Fermi* data (albeit still with non-negligible residuals) while allowing for a clear physical interpretation of the various components (there are also approaches that aim to succinctly describe the Galactic gamma-ray emission, but not to model its origins). One potential concern, however, is that because this method involves so many additional parameters, and the choice of initial templates, which templates are allowed to vary on a pixel-to-pixel basis, and the regularization prescription are all somewhat ad hoc, it is far from clear that this is a uniquely good solution. In other words, it seems plausible that the best-fit SkyFACT model might be

mis-attributing emission that is truly associated with cosmic rays hitting the gas to the GCE, or vice versa.

Nonetheless, SkyFACT (at least as currently implemented) prefers a stellar-bulge-like morphology for the GCE [212], and the authors show in an appendix that this preference arises from the extra degrees of freedom allowed for the Galactic diffuse emission: prior to this freedom being added, a spherical morphology is generally preferred. This means there is at least an existence proof of a model for the *Fermi* sky, with a stellar-bulge-shaped GCE, that fits the data much better than out-of-the-box models and does not obviously conflict with other constraints (e.g. on the gas column density). This is not necessarily a unique solution – but while it may still be possible (for example) to obtain an equally well-fitting description with a spherically symmetric GCE, there is not yet (to the author's knowledge) an existence proof of such a model.

This seems to indicate that it will be very challenging to *exclude* a stellar-bulge-shaped GCE with current data, even if such a GCE morphology appears to be disfavored in a given set of GALPROP-based models – any apparent exclusion could simply be due to not fully accounting for uncertainties in the gas distribution. However, because the difference between a spherical GCE and a bulge-shaped GCE does appear to be at a level that can be absorbed into uncertainties in current interstellar gas maps (possibly combined with uncertainties in other aspects of the background modeling), it also seems somewhat premature to discard the possibility of non-bulge-like morphologies. Thus while the preference for a bulge morphology in the studies of Refs. [212–214] does provide some evidence in favor of a stellar origin, it is (in the view of the author of these notes) hard to argue that it constitutes definitive proof; it would be helpful to have additional lines of evidence pointing in the same direction.

## 6.6 Clues from photon statistics

Another way to distinguish between the various hypotheses is to examine the *clumpiness* of the photons. If the signal originates from DM annihilation or an outflow, we would expect to observe a fairly smooth spatial distribution of flux. In the pulsar case, we might instead see many "hot spots" scattered over a fainter background. This general claim can be made quantitative by considering the differing photon statistics in these two cases; the variance is larger for a given mean when point sources are present, and this can be captured in a modification to the likelihood function, even if the precise locations of the point sources are not known.

### 6.6.1 Methods

As a simple example of the required modification to the likelihood in the presence of point sources, consider a scenario where 10 photons per pixel are expected, in some region of the sky. What is the probability of finding 0 photons? 12 photons? 100 photons?

In a case where only diffuse emission is present, Poissonian statistics are valid; the probability of observing exactly 12 photons is $P(12) = 10^{12}e^{-10}/12! \sim 0.1$, and likewise $P(0) \sim 5 \times 10^{-5}$, $P(100) \sim 5 \times 10^{-63}$. But suppose instead we are told that the source of emission is a population of rare sources, each of which produces 100 photons on average, but with only 0.1 sources expected per pixel. Poissonian statistics now cannot be applied, as observing a photon from a given pixel tells us that there is a source present there, and increases the probability of seeing another photon from the same pixel – the events are no longer independent. The mean number of photons per pixel remains the same, but now the probability of seeing zero photons is $P(0) \sim 0.9$ (neglecting the possibility that a source is present but fluctuates down to 0 photons from the 100 expected), $P(12) \sim 0.1 \times 100^{12}e^{-100}/12! \sim 10^{-29}$,

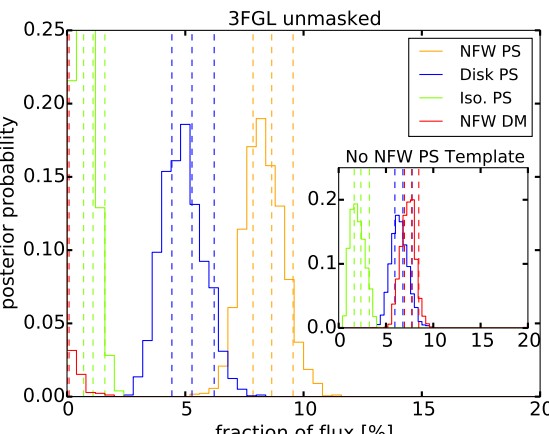

Figure 17: Posteriors for the flux fraction within $10°$ of the Galactic Center with $|b| \geq 2°$ arising from the separate point source components, with known point sources unmasked. The inset shows the result of removing the NFW PS template from the fit. Dashed vertical lines indicate the 16th, 50th, and 84th percentiles. Reproduced from Ref. [235].

and $P(100) \sim 4 \times 10^{-3}$ (in the latter two cases, I have neglected terms arising from the case where multiple sources are present in a pixel, as this is rare).

Thus the expected *distribution* of the number of photons is very different, even though the mean is the same, between the two cases. In the first case, seeing 12 photons is quite likely, but in the second case it is essentially impossible, since this would require a source to be present but to produce only 12 photons when 100 are expected. Likewise, observing 100 photons will never happen in the first case, but is quite plausible in the second, if there are a few hundred pixels with these properties.

In the template fitting method discussed earlier, each template was assumed to possess Poissonian statistics. We can now extend this method to *non-Poissonian* template fitting (described in detail in Ref. [235], based on earlier work by Refs. [236, 237]), and thus include templates corresponding to populations of (potentially unresolved) sources. The overall spatial distribution of the sources is specified as previously, but now the probability of observing a certain number of photons, given a model for the total number of photons, is determined via non-Poissonian statistics (appropriate to a point source population, as above) rather than Poissonian. We model the point-source population as a combination of isotropic extragalactic sources, Galactic sources tracing the disk of the Milky Way, and (optionally) point sources following the spatial distribution of the GeV excess; the diffuse emission is modeled as a combination of the *Fermi* p6v11 diffuse model, the *Fermi* Bubbles, the diffuse isotropic gamma-ray background, and a DM-like template following the spatial distribution of the GeV excess.

As in the example above, the non-Poissonian probability of observing a certain number of photons also depends on the properties of the source population: specifically, the number of sources as a function of their brightness, the *source count function*. For each non-Poissonian template, we can add extra parameters to describe the source count function, and fit for these parameters just as we fit for the normalizations of the templates. As a default, we treat the source count function as a broken power law described by three parameters; the indices above and below the break, and the flux at which the break occurs.

We can then perform fits of two models, one including a template for point sources tracing the GeV excess (labeled "NFW PS"), and one without this template. In both cases we include the smooth "NFW DM" template with morphology characteristic of the GeV excess.

This non-Poissonian template fitting (NPTF) approach is not the only way to characterize

sub-threshold point source populations in the inner Galaxy; some similar approaches have been discussed in Refs. [238–240]. It is also possible to simply search directly for faint hotspots, e.g. using wavelet methods [241]. The NPTF has the advantage of being available as a public software package NPTFit [242], with worked examples. Similar photon statistics (or neutrino statistics) techniques have been used in other contexts in DM indirect detection and particle astrophysics, e.g. to search for point sources in IceCube neutrino data [243], and to determine the composition of the extragalactic gamma-ray background [244–246].

### 6.6.2 Initial results and systematic issues

In 2016, using the NPTF method described above, my collaborators and I found [235] that if the "NFW PS" template is absent, the "NFW DM" template absorbed the GeV excess, as previously. But when the "NFW PS" template was added to the fit, then it absorbed the full excess, driving the "NFW DM" template to zero, as shown in Fig. 17. This suggested the GCE was better explained by a population of point sources than by smooth, diffuse emission, whether from DM or from an outflow of cosmic rays. Simultaneously, Ref. [241] presented a search for faint point sources in the inner Galaxy using wavelet methods, and inferred a significant point source population capable of explaining the full GCE.

However, since 2016, the interpretation of both of these results has been questioned. A follow-up wavelet analysis [247] found that while the authors could indeed identify many faint point sources in the inner Galaxy, almost all the wavelet peaks they found were associated with sources that have now been catalogued by the *Fermi*-LAT Collaboration [248], and a large fraction of these sources could be excluded as possible contributors to the GCE (e.g. because they are extragalactic). Furthermore, masking these sources does not significantly reduce the flux of the GCE as inferred from a template analysis, implying their total contribution to the GCE must be less than $\sim 10-20\%$ (although fainter sources could still explain the bulk of the GCE).

In 2019, Dr. Rebecca Leane and I [249] demonstrated that the original NPTF analysis [235] suffered from undiagnosed systematic errors. In particular, we showed that if the normalization of the "NFW DM" component was allowed to run negative (which is unphysical), the best fit was a case where the normalization of "NFW DM" was minus 5× the standard GCE normalization, and the normalization of "NFW PS" was 6× the standard GCE normalization. In other words, the fit wanted to include far more point sources than could physically be accommodated. Consequently, when we tried adding a simulated DM signal to the real data, the fit failed to recover it until its normalization was more than $5 \times$ larger than the true GCE. At lower normalizations, the fit simply reduced the degree to which the "NFW DM" template was reconstructed as negative; if a prior enforced a physical positive normalization, the fit returned a zero normalization for "NFW DM", as in the original NPTF analysis, and increased the inferred normalization of the "NFW PS" component.

This behavior implied mismodeling in at least some components of our background model – and raised the question of whether this modeling error could be severe enough to mis-attribute a fully smooth GCE to point sources. This was a real concern, given that it could apparently mis-attribute a smooth signal 5× larger than the GCE to point sources.

Ref. [250] demonstrated that this failure of the signal injection test could be explained by mis-modeling of the Galactic diffuse emission. Specifically, using an updated model for the Galactic diffuse emission (based on Ref. [214]), they were able to (1) remove the injection test failure in the real data, and (2) demonstrate that simulating the Galactic diffuse emission with this updated model, but performing the fit assuming the older diffuse model used in Ref. [235], could lead to a bias to the "NFW DM" component similar to what we had observed in Ref. [249]. With this updated model, Ref. [250] still found evidence for point sources in the GCE, but at much reduced significance (Bayes factor of $10^{3.4}$).

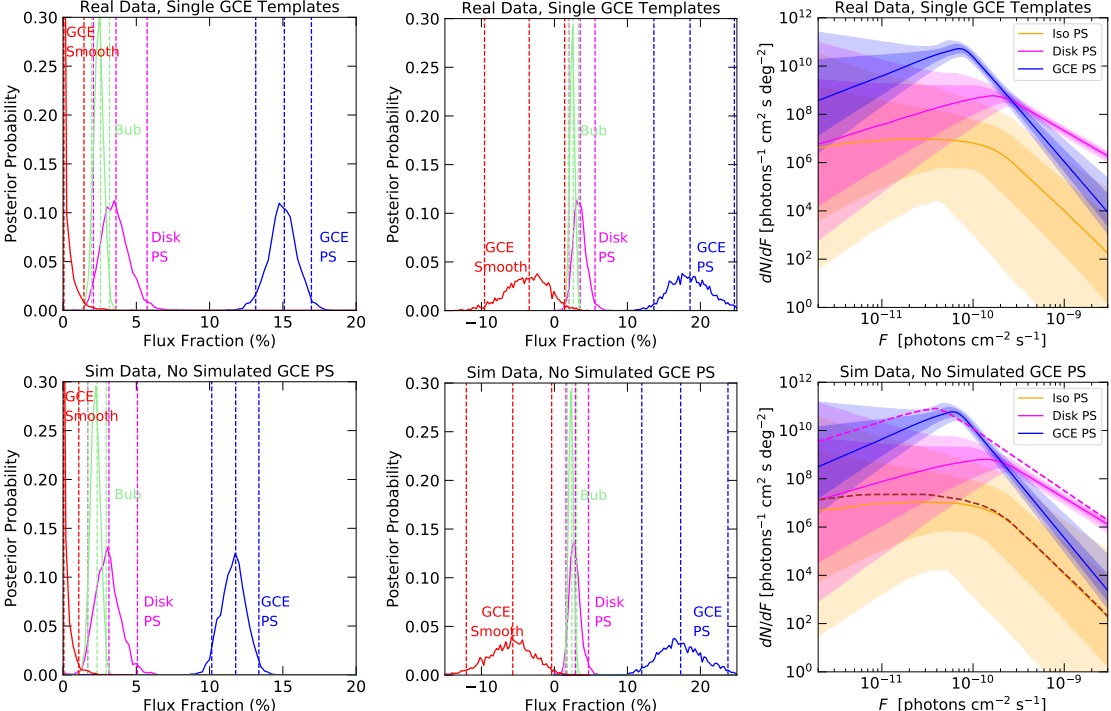

Figure 18: Comparison of real (*top row*) and simulated (*bottom row*) data; in all cases the analyses used symmetric GCE templates (smooth and point sources). The simulated dataset contains a smooth asymmetric GCE and no GCE point sources. **Left column:** Flux posteriors for various templates in the fit where the GCE Smooth component is constrained to have positive coefficient. **Middle column:** Flux posteriors for various templates in the fit where the GCE Smooth component is allowed to float to negative values. **Right column:** SCF corresponding to the left column. The dashed lines on the bottom SCF plot are the simulated SCF for Disk PS (pink) and Iso PS (brown). Reproduced from Ref. [251].

However, in simultaneous work, Dr. Leane and I identified another possible source of systematic bias [251, 252]. We examined a smaller region of 10° radius surrounding the GCE, where the failure of the injection test was less severe, and showed that giving the GCE spatial template one additional degree of freedom – specifically, allowing for north-south asymmetry – completely removed what had appeared to be very strong evidence for GCE point sources, with the fit instead preferring to explain the GCE as asymmetric smooth emission. We then tried simulating a smooth asymmetric GCE and performing the fit assuming symmetric GCE templates ("NFW DM" and "NFW PS"). We found that even when all other templates were perfectly correct, the fit drove the "NFW DM" component to zero and inferred the presence of a "NFW PS" component at high significance. Furthermore, the flux distribution of the (spurious) inferred point sources was very similar to distributions previously inferred in NPTF studies on real data. Fig. 18 illustrates these results.

The GCE asymmetry turned out to depend on both the background model and the region of interest over which the fit was performed, and so we do not believe it is physical – although a confirmed detection of a smooth asymmetric GCE would actually be very interesting, and could reinvigorate hypotheses involving cosmic-ray outflows from the Galactic Center. Instead, we argue that when performing NPTF analyses on the GCE, it is important to ensure the templates have sufficient freedom (some initial work along these lines has been done in Ref. [253]), as a poorly-fitting model can be interpreted as evidence for a highly-significant, convincing-looking

point source population, even when no GCE-associated sources are present at all.

In summary, the GCE could certainly still be comprised of point sources, but previous NPTF-based claims to detect a near-threshold population of point sources at high significance were likely too strong; the inferred source population may be spurious.

## 6.7   Where next?

Efforts continue to characterize the GCE in gamma rays, in particular with regard to its photon-count statistics. As one example, machine learning methods have been developed to try to separate smooth emission from point sources, although so far the results are somewhat inconclusive [254, 255].

If the true solution is a new pulsar population, then such a population could potentially be detected at other frequencies, e.g. by radio or X-ray telescopes. Prospects for detection with the MeerKAT or SKA radio telescopes have been studied in Ref. [256], and an initial study for X-rays has been performed in Ref. [257]. If the true solution is DM or a diffuse signal from cosmic rays, it may be more difficult to establish a fully convincing case – although the AMS-02 antiproton excess discussed in the previous lecture is (if real) a possible counterpart signal; there is a claimed possible counterpart gamma-ray signal from M31 [258, 259]; and given the cross-section needed to explain the GCE, the GAPS experiment [260] could potentially detect anti-deuterons for a confirmation in a low-background channel.

# Acknowledgements

These lecture notes were written for the 2021 Les Houches Summer School on Dark Matter, and adapted and extended from lectures presented at TASI 2016: Anticipating the Next Discoveries in Particle Physics [261]. T.R.S thanks the organizers of the Summer School – Marco Cirelli, Babette Dobrich, and Jure Zupan – for giving her the opportunity to attend and lecture at the School, and the students attending the school for their excellent questions and comments. T.R.S also thanks Joanna Berteaud and the anonymous reviewers for their careful reading of the manuscript and valuable comments. T.R.S's work is conducted with support from the U.S. Department of Energy under grant Contract Number DE-SC00012567.

# A   Review of key results from cosmology

## A.1   The metric and the Friedmann equation

In this section we will briefly review some key results from cosmology that are used in these lectures. We will assume throughout that the cosmos is described by a Friedmann-Lemaître-Robertson-Walker (FLRW) metric,

$$ds^2 = -dt^2 + a(t)^2 \left[ \frac{dr^2}{1 - kr^2} + r^2 d\Omega^2 \right], \tag{19}$$

where $k = 0, \pm 1$ and $a(t)$ is the scale factor describing the overall expansion of the cosmos. The stress-energy tensor corresponding to this metric is that of a perfect fluid with energy density $\rho$ and pressure $P = w\rho$, where $w$ is the equation-of-state parameter.

Applying the Einstein equations of general relativity to this metric and stress-energy tensor yields the Friedmann equation for the evolution of the scale factor $a(t)$,

$$H^2 \equiv \left( \frac{1}{a(t)} \frac{da(t)}{dt} \right)^2 = \frac{8\pi G}{3} \rho - \frac{k}{a(t)^2}. \tag{20}$$

From stress-energy conservation, we can further derive that the energy density $\rho$ of a perfect fluid evolves as $\rho \propto a^{-3(1+w)}$.

We typically model the contents of the universe as a sum of three types of perfect fluid:

- Radiation: $w = 1/3$, $\rho = 3P \propto a^{-4}$. Physically, the number density of a radiation field is diluted as $a^{-3}$ as the volume of the relevant region of the universe expands, but the expansion also stretches the wavelength of the radiation, causing the energy of individual photons (or other radiation quanta) to scale as $a^{-1}$. Thus overall the energy density in radiation scales as $a^{-4}$.

- Non-relativistic matter: $w = 0$, $\rho \propto a^{-3}$. For pressureless matter, the energy of each particle is dominated by its mass, and the overall energy density simply scales like the number density, inversely with the expanding volume.

- Dark energy: $w = -1$, $\rho \propto a^0$. Dark energy is not yet fully understood, but a component with $w$ very close to $-1$ appears to be experimentally required to explain observations. For example, Ref. [262] measures the dark energy equation of state parameter to be $w = -1.01 \pm 0.06$.

Because of these different scalings with $a$, we expect that early in the universe's expansion, when $a$ was much smaller, radiation dominated the energy density of the cosmos $\rho$; at later times, matter came to dominate; and most recently we entered the epoch of dark energy domination. Under the assumption that these are the only contributions to $\rho$, we can write:

$$\rho = \rho_{m,0}\left(\frac{a(t)}{a_0}\right)^{-3} + \rho_{r,0}\left(\frac{a(t)}{a_0}\right)^{-4} + \rho_{\Lambda,0}, \tag{21}$$

where $\rho_{m/r/\Lambda,0}$ represents the present-day energy density in matter, radiation and dark energy respectively, and $a_0$ is the present-day scale factor. Then the Friedmann equation becomes,

$$H^2 = \frac{8\pi G}{3}\left[\rho_{m,0}\left(\frac{a(t)}{a_0}\right)^{-3} + \rho_{r,0}\left(\frac{a(t)}{a_0}\right)^{-4} + \rho_{\Lambda,0} - \left(\frac{a_0}{a(t)}\right)^2 \frac{3k}{8\pi G a_0^2}\right]. \tag{22}$$

Let us define the "critical density" $\rho_c = 3H^2/8\pi G$, $\Omega_X = \rho_{X,0}/\rho_{c,0}$ for $X = m, r, \Lambda$, and $\Omega_k = -k/(a_0^2 H_0^2)$, where $H_0$ is the Hubble parameter at the present time. Then we can write:

$$\frac{H^2}{H_0^2} = \Omega_m\left(\frac{a(t)}{a_0}\right)^{-3} + \Omega_r\left(\frac{a(t)}{a_0}\right)^{-4} + \Omega_\Lambda + \Omega_k\left(\frac{a(t)}{a_0}\right)^{-2}. \tag{23}$$

Current values for the $\Omega$ parameters from observations of the CMB and baryon acoustic oscillations are [61]: $\Omega_m = 0.3111 \pm 0.0056$, $\Omega_\Lambda = 0.6889 \pm 0.0056$, and $\Omega_K = 0.001 \pm 0.002$. The contribution to $\Omega_r$ from photons is $\Omega_\gamma = 5.38 \pm 0.15 \times 10^{-5}$ [263]; at temperatures when neutrinos are relativistic, they would contribute an extra factor $1 + 7/8 N_\nu (4/11)^{4/3}$, where $N_\nu$ denotes the effective number of neutrino species and is 3.046 for SM neutrinos. Thus for the SM this additional factor would be 1.69, for a total $\Omega_r \approx 9.1 \times 10^{-5}$.

## A.2 Redshifting

All particles propagating through the universe on geodesic (free particle) trajectories have their 3-momenta (as measured by a comoving observer) redshifted with the expansion of the universe, i.e. their momenta are reduced by a factor $a(t_{\text{emission}})/a(t)$ at some time $t > t_{\text{emission}}$. For relativistic particles, such as photons, the magnitude of their energy equals that of their 3-momentum, so their energy (and temperature) is inversely proportional to $a(t)$. For non-relativistic particles, their 3-momenta are given by $mv$, and so it is their velocity that redshifts

SciPost Phys. Lect. Notes 53 (2022)

inversely with $a(t)$; consequently, their kinetic energy (proportional to $v^2$) and temperature is proportional to $a(t)^{-2}$. As a result, the temperature of free-streaming matter drops faster than that of radiation due to the expansion of the cosmos, if neither species is heated or cooled by interactions.

We define the redshift $z$ by $1 + z = a_0/a(t)$, where $a_0$ is the scale factor today; thus $z = 0$ today, and $z = 9$ corresponds to the time when the scale factor was $1/10$ of its present size. In the absence of energy/entropy injection, the temperature of the photon bath $T$ scales as $1 + z$, and when performing back-of-the-envelope estimates we will often treat them as proportional.

## A.3  Behavior of the early cosmos

After an early period of radiation domination, matter domination would have ensued when:

$$\Omega_m \left( \frac{a(t_{\text{eq}})}{a_0} \right)^{-3} = \Omega_r \left( \frac{a(t_{\text{eq}})}{a_0} \right)^{-4}, \tag{24}$$

i.e. $1 + z_{\text{eq}} = a_0/a(t_{\text{eq}}) = \Omega_m/\Omega_r \approx 3400$ (a careful calculation yields $z_{\text{eq}} = 3402 \pm 26$ [263]). The present-day temperature of the CMB is 2.7255 K (e.g. [263]), corresponding to an energy of $2.3 \times 10^{-4}$ eV; thus the end of the radiation-dominated epoch corresponds to a temperature around 0.8 eV.

At all higher temperatures, the universe was presumably radiation-dominated after the end of inflation, unless there was another component that decayed away in the interim. Consequently, transitions in the early universe characterized by energy scales above 1 eV are usually assumed to have occurred during radiation domination.

During radiation domination, the energy density of the universe scales as $T^4$ where $T$ is the temperature of the photon bath, and so we can write $H^2 \approx (8\pi G/3)\rho \sim T^4/m_{\text{Planck}}^2$, where $m_{\text{Planck}}$ is the Planck mass (in the last term, we are ignoring all $\mathcal{O}(1)$ numerical prefactors). Thus a quick estimate for $H$ during radiation domination is $H \sim T^2/m_{\text{Planck}}$. The relation between redshift and time is determined by $(da/dt)^2 \propto a^2 a^{-4}$, i.e. $da/dt \propto a^{-1}$, which yields $a \propto t^{1/2}$, or $1 + z \propto t^{-1/2}$.

During matter domination, we instead have $H \propto a^{-3/2}$ and $t \propto a^{-2/3}$, and during dark energy domination, $H$ is constant and $a$ grows exponentially with time.

## A.4  A brief cosmic timeline

- Inflation: the "Big Bang", an accelerated period of expansion at the beginning of the observable universe, terminated by decay of the inflaton field and reheating. Must have completed prior to Big Bang nucleosynthesis (observationally); typically assumed to occur at very high temperatures / redshifts.

- Electroweak symmetry breaking: occurs around $T \sim 160$ GeV, the Higgs field acquires a non-zero vacuum expectation value, particles that interact with the Higgs become massive at this time. In the SM this is a smooth crossover; in extensions to the SM, it can be a first-order phase transition.

- Quark-hadron transition: occurs around $T \sim 100 - 300\, MeV$, when quarks and gluons bind into hadrons. In the SM this is a smooth crossover. At lower temperatures, the appropriate hadronic degrees of freedom are mesons and baryons rather than quarks and gluons.

- Neutrino decoupling: occurs around $T \sim 1 - 3$ MeV (at slightly different temperatures for the different neutrino species), when weak interactions become inefficient compared to the expansion timescale.

- Big Bang nucleosynthesis: begins when the temperature of the universe drops below the typical binding energies of the lightest nuclei, around $T \sim 1$ MeV. Leads to production of light nuclei; measurements of the abundances of these nuclei constitute the earliest direct observational probe of cosmological history.

- Matter-radiation equality: as discussed above, occurs around $T \sim 1$ eV.

- Recombination: when the hydrogen gas goes from being almost completely ionized to almost completely neutral, as the photon bath no longer has enough energy to efficiently dissociate hydrogen atoms. Occurs around $T \sim 0.2$ eV ($z \sim 1000$). Corresponds to the last scattering surface of CMB photons, since once the universe becomes near-neutral it also becomes transparent to the CMB photons.

- Reionization: due to radiation from stars, the gas becomes almost fully ionized again. Occurs between redshift 6-10, corresponding to $T \sim 10^{-3}$ eV.

## B  Review of key results on dark matter freeze-out

Earlier lectures covered a number of production mechanisms for DM. The thermal freeze-out scenario in particular is often employed to motivate indirect searches for DM annihilation, so let us recap the key parametric results here.

Let us focus on the simplest case, where a single DM species annihilates to SM particles with velocity-independent $\sigma v_{\mathrm{rel}}$. What does imposing the measured present-day relic density imply for annihilation cross sections?

In the case with no annihilation, the number of DM particles in a comoving volume would remain constant; if $n$ is the physical (not comoving) DM number density, we would have $\frac{d}{dt}(na^3) = 0$, where $a$ is the scale factor. Expanding this out, we obtain $dn/dt + 3(\dot{a}/a)n = 0$, i.e. $dn/dt + 3Hn = 0$ where $H = \dot{a}/a$ is the Hubble parameter.

In the presence of annihilation, the number density is additionally depleted, yielding:

$$\frac{dn}{dt} + 3Hn = -\frac{n^2}{2}\langle \sigma v_{\mathrm{rel}} \rangle \times 2, \tag{25}$$

for indistinguishable annihilating particles; the second factor of 2 occurs because each annihilation removes two particles, and the use of $\langle \sigma v_{\mathrm{rel}} \rangle$ rather than $\sigma v_{\mathrm{rel}}$ indicates we are averaging over the DM velocity distribution. However, in the presence of annihilation to any set of non-DM particles, the inverse reaction – non-DM particles colliding to produce DM particles – can also occur in principle. Thus we can write:

$$\frac{dn}{dt} + 3Hn = -n^2 \langle \sigma v_{\mathrm{rel}} \rangle + \text{contribution from DM production.} \tag{26}$$

If the DM particles and their annihilation products are in chemical equilibrium, then the contributions from the DM depletion and production processes should cancel out. Thus we can write the DM-production term as $n_{\mathrm{eq}}^2 \langle \sigma v_{\mathrm{rel}} \rangle$, where $n_{\mathrm{eq}}$ is the number density of the DM when it is in chemical equilibrium with its annihilation products, so overall we have:

$$\frac{dn}{dt} + 3Hn = (n_{\mathrm{eq}}^2 - n^2)\langle \sigma v_{\mathrm{rel}} \rangle. \tag{27}$$

If the DM is in equilibrium with the SM thermal bath, its equilibrium number density is given by $n_{\mathrm{eq}} \sim (m_{\mathrm{DM}} T)^{3/2} e^{-m_{\mathrm{DM}}/T}$ when $T \ll m_{\mathrm{DM}}$, and $n_{\mathrm{eq}} \sim T^3$ where $T \gg m_{\mathrm{DM}}$.

When the annihilation rate goes to zero, $n$ evolves as $1/a^3$; when the annihilation rate is large, $n$ will be forced close to $n_{eq}$. The crossover between the two regimes occurs when $\langle \sigma v_{rel} \rangle n^2 \sim Hn$, i.e. $n \langle \sigma v_{rel} \rangle \sim H$. Thus the DM density diverges from its equilibrium value, approaching the constant comoving density that we measure at late times, when the Hubble expansion time becomes comparable to the time needed for a given DM particle to annihilate; we refer to this point as "freezeout". Note that for a constant comoving DM density, $n$ scales as $a^{-3}$ while $H$ scales as $a^{-2}$ during radiation domination and $a^{-1.5}$ during matter domination; thus prior to freezeout $\langle \sigma v_{rel} \rangle$ exceeds $H$ and after freezeout it becomes increasingly small compared to $H$.

(Note that decays do not "freeze out" or "decouple" in this sense, because the relevant rate comparison is $n\Gamma$ to $nH$, and $\Gamma$ is fixed with respect to $a$ while $H$ is decreasing as $a$ increases. Instead, decays are initially inefficient relative to $H$, and become efficient at late times – they go from decoupled $\rightarrow$ coupled with increasing time, rather than coupled $\rightarrow$ decoupled like annihilation processes. Annihilations that are strongly enhanced at low velocities may remain coupled, or re-couple after a period of decoupling.)

A precise calculation of the eventual DM abundance requires numerically solving the evolution equation (eq. 27), accounting for the non-trivial temperature evolution of the SM thermal bath when the number of relativistic degrees of freedom is changing. An up-to-date treatment is given in Ref. [66].

However, a simple estimate can be performed by neglecting these effects, and assuming that freezeout is abrupt and occurs when $H = n_{eq} \langle \sigma v_{rel} \rangle$, that $n$ tracks $n_{eq}$ up to this point, and that after this point the DM number density evolves proportionally to $a^{-3}$. We denote the temperature of the universe at freezeout by $T_f$; we also define the new variable $x \equiv m_{DM}/T$, and write $x_f \equiv m_{DM}/T_f$. Thus the late-time DM number density is given by $n_{today} \approx n_{eq}(T_f) a(T_f)^3 / (a_{today})^3$.

There are broadly two cases to be considered; in the first case, the DM is a "hot relic", and freezes out while still highly relativistic. In this case $n_{eq}(T_f) a(T_f)^3$ is determined entirely by the number of degrees of freedom of the DM, and is almost independent of the freezeout temperature. Consequently, the late-time DM number density is also largely independent of the details of freezeout, and should be comparable to the number density of photons, since the number density of the DM was originally that of a relativistic species and it has only been diluted by the cosmic expansion, not by any other number-changing effects. (This conclusion may be evaded if there are large changes in the number of relativistic degrees of freedom coupled to DM and/or the SM between freezeout and the present day, that affect the two sectors differently, e.g. Ref. [264]). This would suggest a DM mass around the eV scale, since the photon abundance is roughly $2 \times 10^9 \times$ larger than the baryon abundance, the energy density in baryons is comparable to that in DM, and the mass of a proton is 1 GeV. In turn, this mass scale implies that the DM would be relativistic during the epoch relevant to structure formation, and would thus behave as "hot DM"; a scenario where hot DM constitutes 100% of the DM would lead to dramatic changes to structure formation, and is not consistent with observations (see e.g. Ref. [265], or Ref. [266] for more recent constraints).

In the second case, freezeout occurs when the DM is non-relativistic. This scenario can naturally explain a large depletion in the DM number density relative to the photon number density, since at freezeout, $n_{eq} \sim (m_{DM} T_f)^{3/2} e^{-m_{DM}/T_F}$ is exponentially suppressed.

Our condition for freezeout in this second scenario is thus that $H \sim (m_{DM} T_f)^{3/2} e^{-m_{DM}/T_F} \langle \sigma v_{rel} \rangle$. If we assume that freezeout occurs during the radiation-dominated epoch, $H^2 \propto \rho \propto T^4$, and thus $T \propto H^{1/2} \propto t^{-1/2}$ (ignoring any changes in the number of degrees of freedom contributing to the energy density of the universe). Within our approximations, we can thus write $H = H(T = m_{DM}) x^{-2}$, where as previously $x = m_{DM}/T$.

Our freezeout criterion then becomes:

$$H(x=1)x_f^{-2} \sim (m_{\text{DM}}^2)^{3/2} x_f^{-3/2} e^{-x_f} \langle \sigma v_{\text{rel}} \rangle,$$
$$\Rightarrow e^{-x_f} \sim \frac{x_f^{-1/2} H(x=1)}{m_{\text{DM}}^3 \langle \sigma v_{\text{rel}} \rangle}. \tag{28}$$

Since the exponential scaling with $x_f$ on the LHS is much faster than the power-law scaling on the RHS, as a lowest-order approximation we can write $x_f \sim \ln(m_{\text{DM}}^3 \langle \sigma v_{\text{rel}} \rangle / H(x=1))$. Note that $x_f$ has only a logarithmic dependence on the DM mass and the annihilation cross section.

Ignoring changes in the number of degrees of freedom coupled to the thermal bath, the photon number density and DM number density will both scale as $1/a^3$ after freezeout, so for purposes of our estimate, we can approximate $\rho_{\text{DM}}/n_\gamma = m_{\text{DM}} n_{\text{DM}}/n_\gamma$ at freezeout, and require that this match the observed late-time value. Now by the same reasoning as above, $n_{\text{DM}}(x_f) \sim H(x=1)x_f^{-2}/\langle \sigma v_{\text{rel}} \rangle$, whereas $n_\gamma(x_f) \sim T_f^3 \sim m_{\text{DM}}^3/x_f^3$. Thus we can write:

$$\frac{m_{\text{DM}} n_{\text{DM}}}{n_\gamma}(x_f) \sim \frac{H(T=m_{\text{DM}})}{m_{\text{DM}}^2} \frac{x_f}{\langle \sigma v_{\text{rel}} \rangle}. \tag{29}$$

Since $H \propto T^2$, $H(T = m_{\text{DM}})/m_{\text{DM}}^2$ is approximately independent of $m_{\text{DM}}$; writing $H \sim T^2/m_{\text{Planck}}$, we can write:

$$\frac{\rho_{\text{DM}}}{n_\gamma}(x_f) \sim \frac{1}{m_{\text{Planck}}} \frac{x_f}{\langle \sigma v_{\text{rel}} \rangle}. \tag{30}$$

Since $x_f$ is roughly independent of $m_{\text{DM}}$ at lowest order, we see that fixing $\rho_{\text{DM}}/n_\gamma$ to its measured present-day value will also fix $\langle \sigma v_{\text{rel}} \rangle$, to a value that is approximately independent of the DM mass.

Now let us put in some numbers: the DM mass density is roughly $5\times$ the baryon mass density, or $\sim 5\,\text{GeV} \times n_b \sim 5 \times n_\gamma \times 5 \times 10^{-10}$ GeV, since the baryon-to-photon ratio is $\sim 5 \times 10^{-10}$. Since $m_{\text{Planck}} \sim 10^{19}$ GeV, we obtain:

$$\rho_{\text{DM}}/n_\gamma \sim 3 \times 10^{-9} \text{GeV} \sim \frac{x_f}{\langle \sigma v_{\text{rel}} \rangle} 10^{-19} \text{GeV}^{-1}$$
$$\Rightarrow \langle \sigma v_{\text{rel}} \rangle \sim x_f \times 10^{-10.5} \text{GeV}^{-2}. \tag{31}$$

Since $x_f$ is a log quantity, let us first guess that it is $\mathcal{O}(1)$, to give us an estimate of roughly what mass range will yield the correct cross section; if we take $\langle \sigma v_{\text{rel}} \rangle \sim \alpha_D^2/m_{\text{DM}}^2$, and choose $\alpha_D \sim 0.01$ to be comparable to the electroweak coupling of the SM, we infer that $m_{\text{DM}} \sim 10^3$ GeV should yield roughly the right relic abundance. Armed with this information, we can make a better estimate of $x_f$:

$$x_f \sim \ln(m_{\text{DM}}^3 \langle \sigma v_{\text{rel}} \rangle / H(x=1))$$
$$\sim \ln(m_{\text{DM}} m_{\text{Planck}} \langle \sigma v_{\text{rel}} \rangle)$$
$$\sim \ln(10^{22} \text{GeV}^2 \times 10^{-10.5} \text{GeV}^{-2})$$
$$\sim 25. \tag{32}$$

Substituting this back into our expression for $\langle \sigma v \rangle$ yields $\langle \sigma v_{\text{rel}} \rangle \sim 10^{-9} \text{GeV}^{-2} \sim 10^{-26}$ cm$^3$/s, and a natural mass scale for $m_{\text{DM}}$ around a few hundred GeV.

A more careful calculation (e.g. Ref. [66]) gives $\langle \sigma v_{\text{rel}} \rangle \approx 2 - 3 \times 10^{-26}$ cm$^3$/s, almost independent of the DM mass; this cross section is known as the "thermal relic" cross section.

An alternative way to "plug in the numbers" is to say that the radiation and DM densities must be similar at the epoch of matter-radiation equality (since DM constitutes most of the matter density in the universe). If the temperature at matter-radiation equality is $T_{\mathrm{MRE}}$, then since at MRE $\rho_{\mathrm{DM}} \sim \rho_{\gamma} \sim T_{\mathrm{MRE}}^4$ (here we ignore contributions to the radiation density from the neutrinos, as we only need a first-pass estimate), and parametrically the radiation number density at matter-radiation equality is $\sim T_{\mathrm{MRE}}^3$, we have $\rho_{\mathrm{DM}}/n_{\gamma} \sim T_{\mathrm{MRE}}$. Thus to obtain the right relic density we require:

$$T_{\mathrm{MRE}} \sim \frac{1}{m_{\mathrm{Planck}}} \frac{x_f}{\langle \sigma v_{\mathrm{rel}} \rangle}$$
$$\langle \sigma v_{\mathrm{rel}} \rangle \sim \frac{x_f}{T_{\mathrm{MRE}} m_{\mathrm{Planck}}}. \tag{33}$$

If instead we have $N$ DM particles in the initial state ($N > 2$), then the calculation works very similarly, except that we replace $\langle \sigma v_{\mathrm{rel}} \rangle$ with a rate coefficient we label $\langle \sigma v_{\mathrm{rel}}^{N-1} \rangle$, with units of mass$^{-(2+3(N-2))}$. The freezeout condition becomes $n_{\mathrm{DM}}(x_f)^{N-1} \langle \sigma v_{\mathrm{rel}}^{N-1} \rangle \sim H(x_f) \sim T_f^2/m_{\mathrm{Planck}}$, and hence $\frac{m_{\mathrm{DM}} n_{\mathrm{DM}}}{n_{\gamma}}(x_f) \sim \frac{m_{\mathrm{DM}}}{T_f^3} \sqrt[N-1]{T_f^2/(m_{\mathrm{Planck}} \langle \sigma v_{\mathrm{rel}}^{N-1} \rangle)}$. Equating this to $T_{\mathrm{MRE}}$ as above, we find:

$$\langle \sigma v_{\mathrm{rel}}^{N-1} \rangle \sim \frac{x_f^{(3N-5)} m_{\mathrm{DM}}^{2(2-N)}}{m_{\mathrm{Planck}} T_{\mathrm{MRE}}^{N-1}}. \tag{34}$$

We see that we recover our previous result when $N = 2$, but for $N > 2$ the rate coefficient is no longer approximately independent of the DM mass. Suppose we estimate that the rate coefficient scales parametrically as $\langle \sigma v_{\mathrm{rel}}^{N-1} \rangle \sim \alpha_D^N/m_{\mathrm{DM}}^{2+3(N-2)}$, for some effective coupling $\alpha_D$: this is not guaranteed, but will occur if we assume a tree-level cross section with no p-wave suppression and a similar mass scale for both the DM and any mediators to the SM. Then we can write down a condition on the coupling and mass scale:

$$\alpha_D^N \sim \frac{x_f^{(3N-5)} m_{\mathrm{DM}}^N}{m_{\mathrm{Planck}} T_{\mathrm{MRE}}^{N-1}} \Rightarrow m_{\mathrm{DM}} \sim \alpha_D \frac{\sqrt[N]{m_{\mathrm{Planck}} T_{\mathrm{MRE}}^{N-1}}}{x_f^{(3-5/N)}}. \tag{35}$$

Thus we see the relevant mass scale to obtain the correct relic density is parametrically suppressed for $N > 2$, by extra factors of $T_{\mathrm{MRE}}$ in the geometric mean with the Planck scale.

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
