# Peer review of "Les Houches Lectures on Indirect Detection of Dark Matter"

_SciPost Physics Lecture Notes, doi:SciPost Phys. Lect. Notes 53 (2022)_

## Round 1 · Referee Report · Anonymous (Referee 1) · 2021-10-10

Report

This manuscript consists of a very well written set of lecture notes, based on lectures presented by the author at the Les Houches Summer School on Dark Matter earlier this year. These lecture notes provide an authoritative and pedagogical account of dark matter indirect detection, together with an overview of the status of the field. This topic is of ongoing interest to the research community. The material is correct, logically organized, and presented in a clear and accessible manner. The references are comprehensive. This is an excellent resource for graduate students and others who are new to dark matter indirect detection. This manuscript should certainly be published.

---

## Round 1 · Referee Report · Anonymous (Referee 2) · 2021-10-11

Report

In the manuscript the author presents a review of the current status of indirect searches for dark matter (DM). The manuscript is well written and also well structured with respect to the topic discussed. Thus, I have just few minor comments for the author's attention, which nonetheless I hope can be of help for some improvement of the manuscript.

-- Section 2: since the manuscript is meant for a school and it's pedagogical in nature I think it's worth adding some more plot here and there. For example in section 2.3 plots of the energy spectra of the products (gamma rays or e+e-) of 2body annihilation/decay (of course they can be found somewhere else but it would be useful to have them also here to have a self-contained discussion). Or when Sommerfeld enhancement is discussed, it can be mentioned that plots of the cross-section dependence and resonance with mass are shown later in section 4.2.3 and figure 7.

-- bottom of page 13: I think there is a typo: it says
"Here D(E) is a diffusion coefficient, which we approximate to be independent of energy and time;"
but I guess it's meant to be something like:
"Here D(E) is a diffusion coefficient, which we approximate to be time and space independent"
(the energy dependence is present and discussed afterwards)

-- Eq.14 15 e 16: probably it should be made a bit more clear that eq.14-15, referring to the DM smooth case, represent a somewhat academic case since matter in reality is clustered and this changes the result *dramatically* for the annihilating case (4-5 orders of magnitude). In this respect, it is also worth adding some more references besides ref.75. In particular, ref.75 discusses a special treatment which exploits n-body simulations to calculate the DM annihilation signal, but this kind of
calculation is routinely perfumed in a simplified way using the halo model of structures. For example, a seminal paper in this respect is:
Ullio et al. Phys.Rev.D 66 (2002) 123502, astro-ph/0207125
while something more recent is
Ackermann et al. JCAP 09 (2015) 008 astro-ph/1501.05464

-- in relation to the previous point, in section 4.2.2 I think some more detailed mention should be included regarding the DM limits from the extra-galactic gamma-ray background (EGB) since the extragalactic signal is among the most studied targets together with the Galactic center and dwarfs galaxies.
Moreover among the different techniques which are used to study the EGB there is the non-poissonian template fitting,
(Zechlin et al. Astrophys.J.Suppl. 225 (2016) 2, 18 e-Print: 1512.07190,
Lisanti et al. Astrophys.J. 832 (2016) 2, 117 e-Print: 1606.04101,
Feyereisen et al. JCAP 09 (2015) 027 e-Print: 1506.05118)
which is discussed in detail in section 6.6. Thus it would be nice to mention it also here to show that this a general methodology which finds application in multiple contexts.

-- I see that only some brief mention is devoted to DM indirect detection with neutrinos. Is this because the topic is covered in some other lecture of the school? Since the title says generically "indirect detection" I would have expected this topic to be covered. Otherwise I think the preference to focus more specifically only on gamma-rays and cosmic rays and to leave out neutrinos should be better clarified in the abstract and introduction (eventually maybe also in the title).

---

## Round 1 · Referee Report · Anonymous (Referee 3) · 2021-11-8

Report

I have carefully read through the Les Houches Lectures on Indirect Detection of Dark Matter by Tracy Slatyer. I find the lectures to be well-written, consise and informative. I recommend them for publication. I have only a few minor comments, which may be ignored or implemented based on the preference of the author:

1.) The discussion of the effects of freeze-out on pages 3 and 4 is a very interesting approach – but I think it potentially gives students a mistaken impression that freeze-out temperatures are expected to be O(1 MeV) – where in reality the text is simply discussing a worst-case scenario. It is probably worth adding an extra sentence at the top of page 4 to explain that this scenario is being chosen because it represents the worst possible value, rather than a reasonable ballpark prediction for freeze-out.
2.) On page 12, it may be worth mentioning that certain “hard” final states also avail themselves to similar bump-hunt techniques. The constraints can often be almost as strong so long as the gamma-ray spectrum is peaked over a range that is smaller than the instrumental energy resolution.
3.) On the bottom of page 33, the standard diffusion distance for e+e- pairs is given to be O(100 pc) – but this is a very energy-dependent statement, and it is unclear what energy is being used here. It may also be worth noting the energy scaling for cooling vs. diffusion – which is somewhat counter-intuitive.
4.) On page 34, it is probably worth noting that anti-Helium is sort of an unexpected detection – as many of these techniques were used to search for anti-deuterons, which have not been found.

Otherwise, I believe this lecture is a great addition to the literature, and should be published. I leave all of these comments to the author’s discretion, as I believe the article is already in very good shape.

---

## Round 3 · Author Response

I thank all three referees for their helpful remarks. Changes made in response to referee 2 and 3’s comments are given below.

---

## Round 3 · List of Changes

-At referee 2's request, I have added a plot of the photon/positron/neutrino spectra for various final states. I have also referenced section 4.2.3 and figure 7 in the initial discussion of Sommerfeld enhancement.

-I have corrected a typo relating to the diffusion coefficient pointed out by Referee 2 (and the student volunteer reviewer).

-I have added a footnote where the simplifying assumption of DM homogeneity is discussed, clarifying that this is a preliminary example to make the calculation easier to follow, and that realistically the average annihilation rate at late times will be much higher than expected from the cosmological average density. I have also added a parenthetical comment as a reminder of this point in the lead-in to Eqs. 14-15.

-Regarding referee 2's requests to add further details on DM limits from the extra-galactic gamma-ray background (EGB) and neutrino searches, I have retitled the section on current indirect detection signals to clarify that this is intended as a selection of constraints rather than a comprehensive review of indirect searches. I have added a brief discussion of the EGB constraints as a competitive channel and provided a reference for further reading, and slightly expanded the discussion of neutrino signals from DM annihilation/decay with some additional references. I have added a mention of the EGB applications of non-Poissonian template fitting, and citations to the papers mentioned by the referee, in section 6.6.

-As requested by referee 3, I have added a parenthetical comment and a sentence to the discussion of freezeout on pages 3-4 to emphasize where these numbers are (conservative) bounds, not central-value predictions.

-As requested by referee 3, I have added a brief discussion of "hard" final states with peaked gamma-ray spectra on page 12.

-To address a comment by referee 3, on page 33 in v1 (now page 35) I have added context to clarify that I was discussing the expected extension of TeV gamma-ray halos observed by HAWC, not electron/positron diffusion more generally.

-As requested by referee 3, I have reworded my comments on anti-helium detection slightly to clarify the unexpectedness of this signal.

-I have added the citations suggested by the referees, and a few others suggested by other readers.

---

## Editorial Decision

published